# Indoor Air Quality Dataset with Activities of Daily Living in Low to Middle-income Communities

**Prasenjit Karmakar**
IIT Kharagpur, India
prasenjitkarmakar52282@gmail.com

**Swadhin Pradhan**
Cisco Systems, USA
swadhinjeet88@gmail.com

**Sandip Chakraborty**
IIT Kharagpur, India
sandipc@cse.iitkgp.ac.in

## Abstract

In recent years, indoor air pollution has posed a significant threat to our society, claiming over 3.2 million lives annually. Developing nations, such as India, are most affected since lack of knowledge, inadequate regulation, and outdoor air pollution lead to severe daily exposure to pollutants. However, only a limited number of studies have attempted to understand how indoor air pollution affects developing countries like India. To address this gap, we present spatiotemporal measurements of air quality from 30 indoor sites over six months during summer and winter seasons. The sites are geographically located across four regions of type: rural, suburban, and urban, covering the typical low to middle-income population in India. The dataset[1] contains various types of indoor environments (e.g., studio apartments, classrooms, research laboratories, food canteens, and residential households), and can provide the basis for data-driven learning model research aimed at coping with unique pollution patterns in developing countries. This unique dataset demands advanced data cleaning and imputation techniques for handling missing data due to power failure or network outages during data collection. Furthermore, through a simple speech-to-text application, we provide real-time indoor activity labels annotated by occupants. Therefore, environmentalists and ML enthusiasts can utilize this dataset to understand the complex patterns of the pollutants under different indoor activities, identify recurring sources of pollution, forecast exposure, improve floor plans and room structures of modern indoor designs, develop pollution-aware recommender systems, etc.

## 1 Introduction

**Motivation:** In the last decade, researchers have raised concerns about outdoor air pollution from manufacturing, production [1], textile [2], transport [3], etc. With an increasing awareness about the long-term health impacts of pollutants, governments [4, 5], and NGOs [6] in recent years have shown a great deal of interest in studying our environment in a variety of ways. As per 2023 data [7], developing nations like India, China, Bangladesh, etc., are the most affected because foreign nations often offload their chemical and electronic goods production to developing countries with the availability of cheap labor [8]. Increasing outdoor air pollution [9] in these places also contributes to indoor pollution due to rapid and unplanned urbanization, cramped living conditions, compromised household ventilation, and general ignorance [10, 11]. Furthermore, an average person spends 90%

---

[1]https://github.com/prasenjit52282/dalton-dataset (Access:October 21, 2024)

of his time in indoor spaces (i.e., home and corporate offices, factory floors, etc.) [12]. Therefore, indoor air pollution has a more significant impact on our well-being. In India, several reports [13] have emerged over the last couple of years, highlighting the deadly implications of bad indoor air on our health. Risks increase dramatically for infants, homemakers [14], older adults, and sick populations [15], as they stay almost entirely indoors.

Publicly available indoor air quality datasets are primarily divided into in-situ and survey-based categories. In-situ datasets rely on smart devices (i.e., smart plugs, proximity sensors, HVAC systems, etc.), providing context for ventilation and appliance usage. These datasets mainly focus on parameters like user comfort (i.e., temperature, humidity) [16, 17] and power efficiency [18] of the indoor rather than pollution and ventilation. Survey-based datasets rely on recurring questionnaires to contextualize indoor air pollution patterns. Studies like [19, 20] gather student responses on comfort, mood, and engagement to relate with indoor parameters like $CO_2$, temperature and humidity. These datasets present challenges due to delayed user responses, making real-time alignment of pollution events difficult.

**Our Dataset:** Unlike the studies mentioned above, outdoor air pollution has been extensively studied in developing countries [21, 22, 23]. Whereas indoor air pollution studies are very limited. Notable exceptions include [24], which assesses indoor air quality in single and cross-ventilated Indian households. In another work, [25] identifies emission rates of pollutants like $CO_2$, particulate matter, and volatile organic compounds when using different fuels in five kitchens, and [26], which examines pollution sources in Mongolian settlements. However, such experiments are very small-scale and do not always release the dataset to the research community. Other studies, such as [27, 28, 29, 30, 31], provide scenario-specific datasets from various developing countries.

Identifying a gap in the existing publicly available datasets for indoor pollution analysis, we collected a cross-sectional dataset over various indoor environments, such as studio apartments, canteens, labs, and residential homes, over a six-month period in summer and winter. As the size of the indoor space and the number of rooms vary, we have deployed one or more sensors per site. Moreover, we collected the activity annotations from the occupants over the duration of the field study to understand different indoor pollution dynamics and the influence of activities on the pollutants. In addition, we collected spatial geometry and floor plans of the measurement sites to better understand the impact they have on pollution spread and accumulation.

**Dataset Uniqueness:** This dataset is a crucial contribution towards understanding the pollution dynamics of developing countries like India. Through the exploration of this dataset, ML researchers and architects can predict indoor pollution behavior, identify recurring sources of indoor pollution, and devise policies and recommendations for improving indoor air. We also found that the indoor pollutant dynamics are influenced by various factors like floor plan, wind flow, indoor activities, and ventilation system. Overall, our dataset has the following attributes:

- *Multi-device*: The dataset contains pollution measurements from multiple devices at a particular measurement site with multiple rooms. We have deployed a maximum of 6 devices in a residential household. Therefore, the dataset offers unique observations of the spread patterns of the pollutants throughout the day.

- *Indoor types*: The dataset captures measurements from five different types of indoor locations, namely residential households, studio apartments, food canteens, classrooms, and research labs (total 30 sites), providing a generalized view of indoor pollution dynamics.

- *Frequent pollutants*: The dataset offers readings of indoor temperature and humidity along with eight commonly occurring harmful pollutants (i.e., $CO_2$, VOC, $PM_1$, $PM_{2.5}$, $PM_{10}$, $NO_2$, $C_2H_5OH$, CO) due to indoor activities and events.

- *Human annotations*: During data collection, we adapted human-in-the-loop activity annotation with the help of a simple speech-to-text Android application. Therefore, the dataset contains real-time activity labels and thus provides the necessary indoor context to interpret pollution readings.

- *Multi-city deployment*: We have collected data from four geographical regions in India, covering the typical indoor activities and pollution dynamics of rural, suburban, and urban populations.

- *Dataset duration*: The dataset is collected over six months during the Summer (Week 1 to Week 12) and Winter (Week 13 to Week 25) seasons. Therefore, the dataset captures seasonal changes in pollution dynamics and unique human behaviors specific to temperature and humidity variation.

**Possible Dataset Applications:** This indoor pollution dataset captured in a developing and diverse country like India can be used to develop intelligent indoor services and construct realistic models of pollution and ventilation dynamics for indoor spaces. In general, the dataset can be used in the following applications:

- *Pollution Source Identification and Activity Monitoring:* We observe that various pollution sources (i.e., kitchen, disinfectant, etc.) and occupant's activities generate specific pollution patterns based on the activity and how it is performed. The dataset records many such instances, which can be used to learn these unique relationships and develop models for source detection and activity classification.

- *Analysis of Spreading and Accumulation Patterns in Different Floor Plans:* The dataset can be used to analyze the spreading, accumulation, and trapping behavior of indoor pollutants in different indoor floor plans and room structures.

- *Healthy Home Characterization and Improving Designs of Modern Indoors:* The dataset can be used to identify contributory features and design choices of a household that help cope with pollution accumulation and spread, characterizing the healthiness of the household. Further, modern floor plans and room designs can be improved.

- *Smart Device Control (i.e., AC, Exhaust, Air Purifier, etc.):* The dataset can be used to design intelligent control policies to modulate indoor ventilation through precise actuation of exhaust fans, air conditioners, and air purifiers to improve indoor air.

## 2   Related Work

The publicly available indoor air quality datasets are primarily divided into two categories based on the type of user participation or event labeling: (i) in-situ and (ii) survey-based. Additionally, we outline research studies focused on developing nations later in this section.

### 2.1   In-situ Datasets

In-situ methods do not require user input; instead, they gather information such as the status of the windows (open/closed), the status of the doors (open/closed), or any other equipment (e.g., air conditioners, fans, smart plugs, etc.) statuses (on/off) in order to understand the overall usage of appliances and ventilation. Several studies have followed this strategy since it is agnostic of the occupants and does not require them to actively participate in capturing indoor event context. For instance, [32, 33, 34] have deployed air quality sensors in smart buildings and residential houses along with proximity and light sensors, smart plugs to track occupancy and appliance usage of the buildings. Similarly, there are several works [18, 35, 36] that record air quality measurements along with air conditioning, smart plugs, and lighting loads, as well as other electrical appliance usage to determine building energy performance. However, most of such works [37, 16, 38, 17] are restricted to measuring indoor comfort parameters such as temperature and humidity, entirely ignoring the pollution and ventilation context. Overall, most in-situ studies consider the comfort and energy aspects of indoor environments.

### 2.2   Survey-based Datasets

The survey-based method relies on recurring user surveys at a fixed time interval in order to gather indoor context data, including user comfort, clothing, activities, and events. In recent years, several studies have adapted surveys as a medium of user interactions. For instance, [19, 39] have collected feedback from students about their mood, attention levels, clothing, and classroom engagement to correlate with indoor pollutants and physiological readings in a private school. Similarly, [20] shows how temperature and $CO_2$ levels can influence the comfort and tiredness levels of students in a classroom. Some recent works [40, 41] have analyzed the thermal comfort of the occupants in hostel buildings inside university premises in India. Most of the survey-based studies [41, 40, 42, 43] consider only unreliable self-reported comfort levels of the occupants in terms of temperature. Furthermore, a few studies [20, 19, 44] have considered the impact of pollutants. In summary, due to delayed user responses, survey-based methods have limited utility in measuring indoor pollution due to their difficulty aligning with real-time dynamic events.

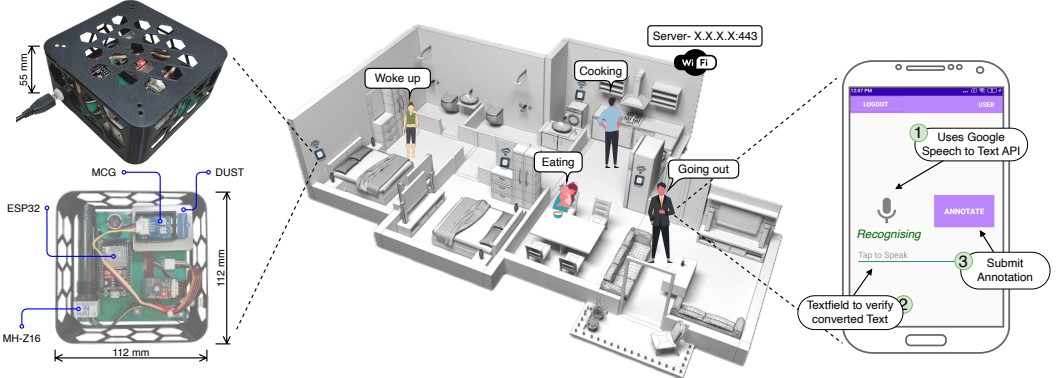

Figure 1: Overview of our extensive field study and data collection with multiple air quality monitors in a typical indoor environment. The scenario shows four *DALTON* sensors deployed in a household that are utilizing the house's WiFi network to send pollutant readings to the cloud. Moreover, the occupants actively participate in the study by providing activity and event context (i.e., cooking, eating, etc.) via the easy-to-use speech-to-text *vocalAnnot* Android application.

## 2.3   Datasets from Developing Countries

Developing countries such as India, China, Africa, and others have undertaken extensive studies of outdoor air pollution due to the involvement of government and non-governmental organizations. As an example, [21] made use of government-deployed air monitoring stations in Durgapur and Delhi to estimate air quality at multiple locations in the city. [22] constructed a mobile sensor network utilizing lower-cost dust sensors mounted on public buses and shared a novel dataset comprised of $PM_{2.5}$ and $PM_{10}$ readings. [23] released a comprehensive dataset collected through the PurpleAir network across a wide range of geographic areas throughout China.

However, a limited number of studies have been conducted in indoor environments. Therefore, there are few datasets available for researchers to understand the intricate dynamics of indoor pollution. For example, [24] investigated air quality in 54 residential households with one-side and cross-ventilated rooms. [26] analyzed data from highly polluted Mongolian ger settlements to determine the sources of particulate matter and how electric stoves could compensate. [27] collected air pollutants such as $CO_2$, ammonia, and volatile organic compounds in a restricted indoor setup with four coarse-grained activity labels such as normal, preparing meal, presence of smoke, cleaning. [28] releases year-long indoor pollutant measurements from 100 air purifiers operating in 18 provinces of China. [29] releases air quality records collected over a one-year period in an indoor travelers' transit area in Brindisi airport, Italy. [30] measured personalized pollution exposure from 82 participants in indoor as well as outdoors with wearable pollution monitors, physiological sensors, and time activity diaries (e.g., car, motorbike, playing, sports, cooking, smoking, etc., totaling 14 activities) for two weeks across summer and winter in Slovenia. [31] releases air quality data from 24 classrooms at schools in Stellenbosch, South Africa, collected for almost a year.

It is important to note that the studies and datasets mentioned above are very scenario-specific or small-scale; in contrast, we present a broader, more comprehensive, and more diverse dataset that we collected across four cities in a developing country (India) in summer and winter, *covering a period of six months of recording air quality data from 30 indoor spaces with real-time activity annotations*.

## 3   Dataset Description

This section explains our sensing setup and the procedure we follow to standardize our data collection across each deployment site. The data is collected across several households and indoor locations over six months. These indoor pollution sensing setups, therefore, had to be affordable, available, and maintainable remotely. In our initial research, we find that the commercial air quality monitors[45, 46, 47, 48, 49] have several shortcomings primarily in terms of remote maintenance and scalability such that they are not suitable for our setup. Thus, we developed a low-cost multi-sensor air

Table 1: Parameters of the dataset and their description.

| Parameters | Description |
|---|---|
| ts | Timestamp (yyyy/mm/dd HH:MM:SS) from the ESP32 MCU after reading sensor values |
| T | Temperature reading of the indoor environment in celsius at time ts |
| H | Humidity reading of the indoor environment in percentage at time ts |
| PMS1 | Less than 1 micron dust particle readings in parts per million (ppm) at time ts |
| PMS2_5 | Less than 2.5 micron dust particle readings in ppm at time ts |
| PMS10 | Less than 10 micron dust particle readings in ppm at time ts |
| CO2 | Carbon dioxide concentration in ppm at time ts |
| NO2 | Nitrogen dioxide concentration in ppm at time ts |
| CO | Carbon monoxide concentration in ppm at time ts |
| VoC | Volatile organic compounds concentration in parts per billion (ppb) at time ts |
| C2H5OH | Ethyl alcohol concentration in ppb at time ts |
| ID | Unique identifier of the deployed *DALTON* sensor |
| Loc | Location of *DALTON* sensor in the indoor environment |
| Customer | The name of the occupant who participated during the sensor deployment in his indoor space |
| Ph | Phone number of the customer for urgent contact. Replaced with XXXX to preserve privacy |

quality monitoring platform that can scale according to the area of indoor space and ensure remote maintenance and debugging capabilities. Moreover, our sensing setup also includes an Android application that allows users to interact with the underlying system by providing activity annotations. These annotations help to uncover the correlations between the sources of pollution and indoor activities being performed. Next, we discuss the sensing setup in detail.

## 3.1 *DALTON* Sensor Module

We have employed an air quality monitoring platform called *DALTON*, which is designed considering low-cost, large-scale field deployment. The module is very compact and easy to deploy, measuring equivalent to a typical lunchbox. It is equipped with multiple research-grade sensors that collectively measure the concentration of indoor pollutants, including Particulate matter ($PM_x$), Nitrogen dioxide ($NO_2$), Ethanol ($C_2H_5OH$), Volatile organic compounds (VOCs), Carbon monoxide (CO), and Carbon dioxide ($CO_2$). Additionally, the module measures temperature (T) and relative humidity (H). The ESP-WROOM-32 chip is utilized as the on-device data processing unit, which consists of a dual-core Xtensa 32-bit LX6 MCU with WiFi capabilities operating at a frequency of 2.4 GHz with HT40 support. The 3D-printed shell features a hollow honeycomb structure that allows unbiased measurement of the pollutants at one sample per second (1 Hz) frequency. Moreover, each *DALTON* sensor is assigned a unique ID and auxiliary information (i.e., location, customer name, etc.) during field deployment, as demonstrated in the Figure 1. The complete list of parameters collected from these sensor modules is described in Table 1.

## 3.2 Annotating Activities of Daily Living

An alert-based Android application has been designed to record specific indoor activities whenever any significant change is detected in pollution concentration. The application is displayed in Figure 1. As shown in the figure, the user must tap the microphone icon to enable speech recognition (i.e., Google's Speech-to-text API) and commence speaking. After accurately converting the speech, the text area is filled with the label text. Before submitting the activity annotation, the user can review and make any necessary edits to the text. Continually annotating using only voice reduces the physical and cognitive work required to report daily activities.

Moreover, we deployed multiple sensor modules in a household to capture the spatiotemporal aspect of pollutants. Individually labeling the data for each sensing module is repetitive and adds to the cognitive burden of the participant. Thus, we associate pollution events with a subset of modules that experience similar trends of pollutants and are placed adjacently. It allows us to identify different spatiotemporal groups of modules within an indoor space and reduce the number of annotation alerts to the participant. On submission, the application triggers a post request with a JSON payload {'Customer':'Name', 'Label':'Activity Label', 'ts':'yyyy-mm-dd HH:MM:SS'} to the IoT backbone of *DALTON* platform where the data is stored.

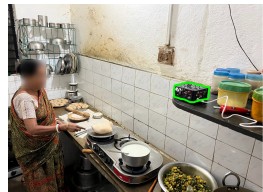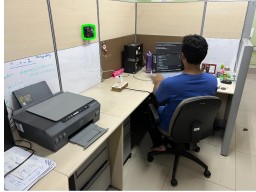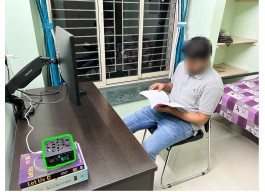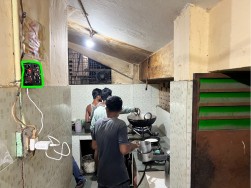

|        (a) Kitchen        |        (b) Research Lab        |        (c) Studio Apartment        |        (d) Food Canteen        |

Figure 2: Deployment images from various indoor environments. The sensor is highlighted in green outline. We have strategically installed at least one sensor in each room (e.g., a typical household can have six rooms). The devices are positioned at a height of 1 meter to 1.5 meters from the ground (i.e., around chest height) based on the availability of standard power outlets to accurately quantify the exposure level for the occupants.

Table 2: Summarization of the economic status, cooking medium, available ventilation, and air conditioning options in different sites in four deployment regions.

| City | | Site | | Occupants | | Ventilation | | | Air Condition | | Cooking Medium | | |
|---|---|---|---|---|---|---|---|---|---|---|---|---|---|
| Name | Type | Site Type | # Sites | Female (%) | Income | Window | Vent-slit | Fan | W | S | LPG | Microwave | Kerosene |
| Bankura | Rural | Household (H1-H13) | 2 | 50 | Low | ✓ | ✓ | ✓ | ✗ | ✗ | ✓ | ✗ | ✓ |
| Durgapur | Suburban |  | 2 | 50 | Middle | ✓ | ✓ | ✓ | ✗ | ✓ | ✓ | ✓ | ✗ |
| Kolkata | Urban |  | 4 | 44 |  | ✓ | ✓ | ✓ | ✓ | ✓ | ✓ | ✓ | ✗ |
|  |  |  | 5 | 60 | Middle | ✓ | ✓ | ✓ | ✓ | ✓ | ✓ | ✓ | ✗ |
| Kharagpur | Suburban | Apartment (A1-A8) | 8 | 33 | Low | ✓ | ✗ | ✓ | ✗ | ✗ |  | – |  |
|  |  | Food Canteen (F1-F2) | 2 | 50 | Middle | ✗ | ✓ | ✓ | ✗ | ✗ | ✓ | ✗ | ✗ |
|  |  | Research Lab (R1-R5) | 5 | 11 | Low | ✗ | ✗ | ✓ | ✓ | ✓ |  | – |  |
|  |  | Classroom (C1-C2) | 2 | – | – | ✗ | ✗ | ✓ | ✗ | ✓ |  |  |  |

### 3.3 Measurement Sites & Demographics of the Occupants

We have collected data from 30 measurement sites in four geographic regions in India for a duration of six months. The data primarily focuses on five types of indoor environments: houses, studio apartments, research labs, food canteens, and classrooms. We have meticulously selected the four regions to capture the usual patterns of indoor pollution dynamics in developing nations. For instance, Bankura is a rural area with unplanned buildings, resulting in densely populated neighborhoods. The houses are naturally ventilated, and individuals are habituated to daily cooking with locally acquired food items and utilizing LPG stoves, firewood, incense sticks, and similar materials. In contrast, Durgapur is an organized industrial city with multiple operational steel and sponge iron industries, leading to substantial pollution in the outdoor environment. The Kolkata is a metropolitan city with a predominantly working population who are accustomed to air conditioners, packaged food ingredients, LPG stoves, and microwave ovens. Finally, Kharagpur is a university town with student apartments, faculty housing, canteens, restaurants, and classroom complexes. The Figure 2 displays a few deployment scenarios (i.e., kitchen, research lab, studio apartment, food canteen) of the *DALTON* sensor during the field study. We summarize our deployment in four regions, their ventilation and air conditioning options, and cooking medium in Table 2.

In total, there were 46 occupants in the measurement sites of the study. Among them, 24 occupants actively participated by providing the annotations for activity and event context of their indoor space using the *vocalAnnot* application as discussed in Section 3.2. For instance, in a household with four occupants, two of them annotated indoor activities and events for everyone. The occupants include college students, university staff, professors, homemakers, canteen owners, etc. Among them, 27 were male, and 19 were female. Table 2 depicts the female participation ratio and the socioeconomic background of the occupants. The majority of the occupants are between 20 and 40 years old. Thus, they are familiar with using Android apps during their everyday routines. Nevertheless, the youngest is a three-year-old child, while the oldest is an 84-year-old adult. Individuals, either too young or too old, do not actively participate in this study, and other family members report their indoor activities.

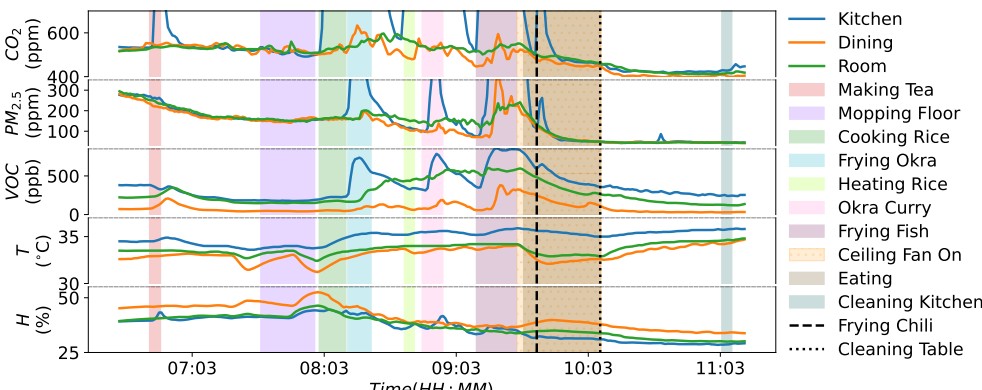

Figure 3: Air quality variation with activities of daily living in the morning time. The figure shows $CO_2$, $PM_{2.5}$, and VOC concentration in the kitchen, adjacent bedroom, and dining while preparing meals for lunch. Long-term frying (e.g., fish) significantly elevates $PM_{2.5}$ and VOC levels that transcend to nearby rooms. Meanwhile, pollutants from boiling, heating, or short-term frying remain contained near the source and do not lead to severe spread. Cleaning and mopping activities increase the relative humidity of the indoors.

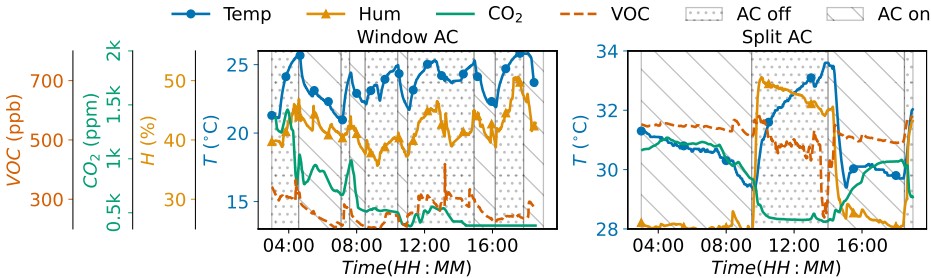

Figure 4: Split air conditioners suffer from compromised ventilation compared to legacy window air conditioners simply to improve power efficiency. The figure clearly depicts the accumulation of VOC and $CO_2$ over time when split AC is on. Meanwhile, we observe consistent ventilation when the window AC is on, achieving healthy air quality with time. The relative humidity is slightly higher for window AC (i.e., within the comfort range, 30–50%).

## 3.4 License and Consent

The dataset is free to download and can be used with GNU Affero General Public License for non-commercial purposes. All participants signed forms consenting to the use of collected pollutant measurements and activity labels for non-commercial research purposes. The participants received $50 per week as remuneration during the field study. The institute's ethical review committee has approved the field study (Order No: IIT/SRIC/DEAN/2023, Dated July 31, 2023). Moreover, we have made significant efforts to anonymize the participants to preserve privacy while providing the necessary information to encourage future research with the dataset.

## 4 Dataset Analysis

**Residential Household:** Pollutants in the indoor environment accumulate and disperse differently depending on the activity. Therefore, various indoor components act as pollution sources during different periods of our daily lives—emissions of indoor pollutants coincide with household activities. As a result, we conclude that the concentrations of $CO_2$ and VOC differ considerably between the dining area, bedroom, and kitchen at various times of the day. For example, the kitchen releases $CO_2$ when cooking occurs, as shown in Figure 3. However, with adequate ventilation, such as exhaust fans and open windows, the $CO_2$ rapidly returns to normal levels as depicted in Figure 5a. In contrast,

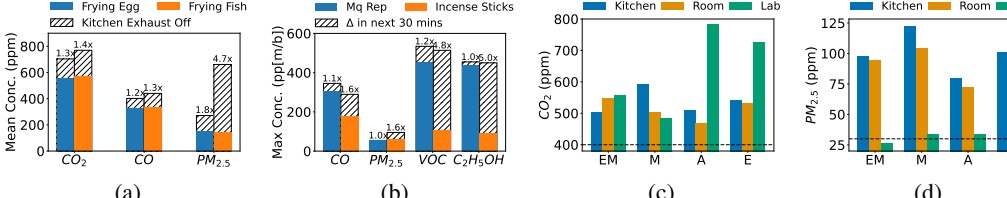

(a)       (b)       (c)       (d)

Figure 5: (a) depicts pollutant accumulation if the kitchen exhaust fan is turned off (i.e., $4.7\times$ $PM_{2.5}$). (b) shows the maximum pollutant increase before and after 30 minutes of using mosquito repellent (Mq Rep) or burning incense sticks (i.e., $4.8\times$ VOC for burning sticks). Further, we have divided the day into Early Morning (EM, 00:00–06:00), Morning (M, 06:00–12:00), Afternoon (A, 12:00-18:00), and Evening (E, 18:00-23:00) hours. (c) shows in the morning, the kitchen has higher $CO_2$, while research labs are impacted in the afternoon and evening due to occupancy. Lastly, (d) shows that $PM_{2.5}$ is predominant in the kitchen and rooms at any time of the day. The dashed line represents healthy pollutant level.

the bedroom experiences elevated $CO_2$ levels at night, surpassing even the kitchen's maximum $CO_2$ levels. This can be primarily attributed to inadequate ventilation as per Figure 4. A similar accumulation pattern of VOC is observed in the kitchen due to leftover food. Conversely, VOC persists for a prolonged duration despite adequate ventilation. This characteristic leads to long-term exposure. Moreover, Figure 5b shows that several daily practices like burning incense sticks or using mosquito repellent influence air quality.

**Classrooms & Research Lab:** Occupancy primarily influences the air quality of academic environments (i.e., classrooms and labs). The concentration of $CO_2$ increases continuously when the lab is occupied but decreases marginally during lunch, dinner, and breaks due to reduced lab activity. The students typically report to the laboratory at approximately 10:00 daily. The accumulation of $CO_2$ continues until 14:00, when the members depart for lunch. Nevertheless, $CO_2$ concentration remains constant due to reduced ventilation with split AC. The pollutant concentration rises in the afternoon and evening due to the peak occupancy as shown in Figure 5c. Further, the $CO_2$ accumulates throughout the evening hours until the dinner break. In brief, individuals' occupancy patterns and activities substantially impact $CO_2$ and other indoor pollutants.

**Kitchens & Food Canteens:** We observe that pollutants are emitted rapidly from any kitchen and canteen environments. Activating the ventilation system or opening the windows can reduce the exposure. However, if the emission rate is slow but ventilation is poor, pollutants get trapped for an extended period since humans are more sensitive to rapid environmental changes. The kitchen observed a sudden peak of $CO_2$ while mostly remaining within the safety threshold ($\leq 1000$ ppm). However, pollutants spread toward interior rooms from the kitchen due to airflow, as per Figure 6.

The dataset captures the general human behavior in the kitchen while choosing to turn on the exhaust fan for ventilation. Even though the concentration of pollutants in the kitchen is significantly reduced after turning on the ventilation, as observed from the collected data, we found that turning on the exhaust fan is conditioned on relatively higher environmental temperature than when the occupants choose not to do so. Thus, we hypothesize that *occupants ignore ventilation when the temperature is comfortable, even though the pollutants start accumulating up to harmful levels*.

## 5  Limitations and Conclusion

Although this is the only large-scale indoor air pollution open-sourced dataset from India with proper annotations, we acknowledge that the dataset has several limitations, including missing labels, discontinuity in measurements from frequent power outages, and short data collection times. The primary limitations are:

- A major limitation is that not all the households and indoor sites have annotations from the users. This is because some participants fail to check their phones for notifications, miss annotations,

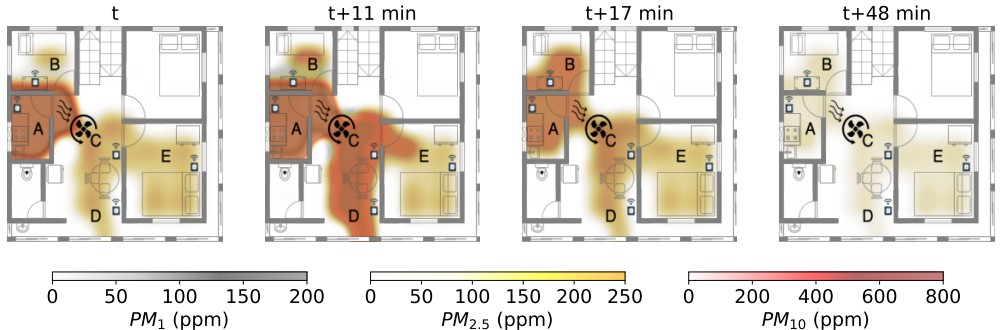

Figure 6: Spread of particulate matter from the kitchen to other interior rooms of the household due to swirling airflow from the dining ceiling fan. The kitchen emits pollutants from $t$ to $t + 11$ minutes. Lighter dust particles $PM_1$ and $PM_{2.5}$ spread more aggressively compared to heavier $PM_{10}$ particles.

or delay providing annotations beyond the specified timestamp. A dataset with such mislabeling scenarios is hard to detect.

• Many of our deployment sites in India do not have a power backup. So, we have experienced several power outages in the measurement sites ranging from a few minutes to several hours. In the post-processing stage, we have to interpolate the readings during the outage in less than 15 minutes.

• Lastly, the dataset contains only six months of measurements from several types of indoor spaces, with the majority of studio apartments limiting the measurement of spread. Additionally, the dataset does not capture indoor dynamics during monsoons and autumns.

To conclude, in this work, we have collected a cross-sectional indoor air quality dataset from low to middle-income households and indoor sites in India. Unlike publicly available data, our dataset captures the indoor pollution dynamics of developing countries. Data was collected over six months across 30 indoor sites, including research labs, canteens, classrooms, studio apartments, and residential properties, with 46 occupants, among which 24 actively participated by providing annotations on indoor activities. Additionally, we include the geometry of the measurement sites, which can help understand the impact of floor plans on pollutant spread. We believe that this dataset can aid environmentalists, architects, and ML researchers in improving indoor designs, addressing pollution challenges, and developing models for predicting indoor pollution patterns.

## 6 Acknowledgement

The authors would like to thank the anonymous reviewers for the constructive comments, which have helped to improve the overall presentation of the paper. The research of the first author is supported by the Prime Minister Research Fellowship (PMRF) in India through grant number IIT/Acad/PMRF/SPRING/2022-23, dated 24 March 2023. The work is also supported by Google's Award for Inclusion Research on Societal Computing 2023 for the project proposal "*AI-Assisted Distributed Collaborative Indoor Pollution Meters: A Case Study, Requirement Analysis, and Low-Cost Healthy Home Solution For Indian Slums.*"

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

# A  Dataset Documentation

We organize the dataset documentation inspired by the template of Datasheets for datasets [52]. The dataset is open-sourced and hosted on GitHub (https://github.com/prasenjit52282/dalton-dataset).

## A.1  Motivation

**For what purpose was the dataset created? Was there a specific task in mind? Was there a specific gap that needed to be filled? Please provide a description.**

The dataset was created to understand the complex dynamics, such as the spread, accumulation, and lingering of pollutants in different room structures and floor plans, specific to low to middle-income households in India. Secondly, the dataset was procured to study recurring pollution sources that occur due to occupants' behaviors and daily activities. The existing studies and datasets are very scenario-specific (i.e., user comfort, power efficiency of buildings, etc.) or small-scale, lacking a generalized view of air quality dynamics. Moreover, indoor air quality datasets are very limited in developing countries. To bridge this gap, we present a broader, more comprehensive, and more diverse dataset that can provide the basis for data-driven learning model research to cope with unique indoor pollution patterns in developing countries.

**Who created the dataset (e.g., which team, research group) and on behalf of which entity (e.g., company, institution, organization)?**

The dataset was created by Prasenjit Karmakar, Swadhin Pradhan, and Sandip Chakraborty. The authors are researchers affiliated with the Indian Institute of Technology Kharagpur (India) and Cisco Systems (USA).

**Who funded the creation of the dataset? If there is an associated grant, please provide the name of the grantor and the grant name and number.**

This work by the authors is partly supported by the Prime Minister's Research Fellowship (PMRF) by the Department of Science and Technology, Government of India, AI4ICPS IIT Kharagpur project grant (Approval Number TRP3RD3223323, dated 22/02/2024). This work is also supported by the Google's Award for Inclusion Research on Societal Computing 2023.

## A.2  Composition

**What do the instances that comprise the dataset represent (e.g., documents, photos, people, countries)? Are there multiple types of instances (e.g., movies, users, and ratings; people and interactions between them; nodes and edges)? Please provide a description.**

There are four types of data in our dataset: (1) Pollutant readings from the air quality sensors, (2) Indoor activity and event annotations from the users, (3) The metadata of sensors, (4) Floor plans of the indoor sites and the location of the sensors that describe the adjacency between the sensors.

**How many instances are there in total (of each type, if appropriate)?**

The dataset presents spatiotemporal air quality measurements from 30 indoor sites (e.g., studio apartments, classrooms, research laboratories, food canteens, and residential households) over six months during the summer and winter seasons (89.1M samples, totaling 13646 hours of air quality data and 3957 activity annotations from 24 participants among total 46 occupants).

**Does the dataset contain all possible instances, or is it a sample (not necessarily random) of instances from a larger set? If the dataset is a sample, then what is the larger set? Is the sample representative of the larger set (e.g., geographic coverage)? If so, please describe how this representativeness was validated/verified.**

Our dataset is a sample of instances from a larger set in both spatial and temporal aspects. In terms of spatial coverage, the larger set includes all types of indoor environments with various room structures and floor plans in India. As for the temporal aspect, the larger set consists of pollutant dynamics for all activities and indoor events. We consider our sample to be representative of the larger set, as we specifically select four geographically distributed regions covering rural, suburban, and urban populations in India. Also, we include six months of recent data to ensure sufficient time coverage.

Moreover, we are still collecting data from different indoor sites, and the dataset is expected to grow in both spatial and temporal aspects.

**What data does each instance consist of? "Raw" data (e.g., unprocessed text or images) or features? In either case, please provide a description.**

The pollution readings are numerical values. The metadata comprises sensor placement and user participant identifiers at each measurement site. Activity annotations are textual descriptions with the activity's starting timestamp. Moreover, the floor plans are images that represent the placement and adjacency of the sensors with respect to the room structure of an indoor space.

**Is there a label or target associated with each instance? If so, please provide a description.**

Yes, the activity and indoor event annotations serve as labels or targets associated with the change in air quality of the indoor environment. These labels are sparse unprocessed textual inputs (e.g., Frying fish, Turning on window AC, etc.) from the participants and describe the activity or indoor event in detail, whenever possible.

**Is any information missing from individual instances? If so, please provide a description, explaining why this information is missing (e.g., because it was unavailable).**

Yes, the sensors are not equipped with power backup; thus, multiple segments of pollutant readings are missing due to power or network outages. Additionally, participants sometimes forgot to respond to the notification from the *vocalAnnot* application, missing the annotation from time to time. Lastly, a few floor plans were left undocumented during data collection.

**Are relationships between individual instances made explicit (e.g., users' movie ratings, social network links)? If so, please describe how these relationships are made explicit.**

Yes, we provide the metadata of the sensor location in each measurement site and an associated floor plan to represent the adjacency relationship between each sensor. Moreover, the activity annotations are timestamped and can be related to the sensor readings.

**Are there recommended data splits (e.g., training, development/validation, testing)? If so, please provide a description of these splits, explaining the rationale behind them.**

Yes, as the dataset comprises temporal measurements, it can be chronologically split into the train, validation, and test sets (widely recommended ratio 6:2:2).

**Are there any errors, sources of noise, or redundancies in the dataset? If so, please provide a description.**

Yes, the sensor readings are never perfectly accurate (i.e., within the reported error margin by the vendor), and there are occasional missing values due to power, network outages, and electrical surges in the deployment site. The processed dataset does not have any redundancy. However, the inherent noise of sensors might be present even after preprocessing (discussed in Appendix A.5). In contrast, the raw dataset files contain multiple replicas to ensure reliable storage primitives.

**Is the dataset self-contained, or does it link to or otherwise rely on external resources (e.g., websites, tweets, other datasets)?**

Yes, it is self-contained.

**Does the dataset contain data that might be considered confidential (e.g., data that is protected by legal privilege or by doctor-patient confidentiality, data that includes the content of individuals' non-public communications)? If so, please provide a description.**

Yes, the dataset contains annotations of participant's daily activities and indoor events, which might be considered confidential in specific scenarios. Therefore, the authors have taken explicit consent from the participants before open-sourcing the dataset. In contrast, air quality monitors are pervasive sensors and do not breach individuals' privacy.

**Does the dataset contain data that, if viewed directly, might be offensive, insulting, threatening, or might otherwise cause anxiety? If so, please describe why.**

No, the dataset does not contain such instances.

Table 3: Overall specifications of the sensing setup.

| System Specification | | | Sensor | | Operational Details | | | | |
|---|---|---|---|---|---|---|---|---|---|
| | | | | | Range | Resolution | Error Margin | Response Time | Operational Temp & RH |
| Microprocessor | | Xtensa®32-bit LX6 Clock 80~240 MHz | DUST [53] | $PM_x$ | 0~500 $\mu g/m^3$ | 1 | ± 10 $\mu g/m^3$ @0~100 $\mu g/m^3$ ± 10% @100~500 $\mu g/m^3$ | ≤10 s | -10~60 °C 0~99% |
| Memory | ROM | 448 KB | | RH | 0~99 % | | ± 2% | | |
| | SRAM | 520 KB | | T | -20~99 °C | 0.1 | ± 0.5 °C | | |
| Connectivity | | Wi-Fi 2.4GHz | MCGS [54] | $NO_2$ | 0.1~10 ppm | | – | ≤30 s | -10~50 °C 0~95% |
| Scan Rate (Hz) | | 1 | | $C_2H_5OH$ | 1~500 ppm | 1 | | | |
| Max Power (W) | | 3.55 | | VOC | | | | | |
| Max Current (mA) | | 760 | | CO | 5-5000 ppm | 0.5 | | ≤10 s | |
| Dimensions(mm) | | 112 × 112 × 55 | MH-Z16 [55] | $CO_2$ | 0~10000 ppm | 1 | ± 100ppm +6%value | ≤30 s | |
| Weight (g) | | 160 | | | | | | | |
| Power Adapter | | DC (5V, 15W) | | | | | | | |

## A.3 Collection Process

**How was the data associated with each instance acquired? Was the data directly observable (e.g., raw text, movie ratings), reported by subjects (e.g., survey responses), or indirectly inferred/derived from other data (e.g., part-of-speech tags, model-based guesses for age or language)?**

We have deployed sensors in each room of an indoor space to measure air pollutant concentration. The occupants actively participate in the data collection process by annotating their daily activities and indoor events using a simple speech-to-text Android application.

**What mechanisms or procedures were used to collect the data (e.g., hardware apparatuses or sensors, manual human curation, software programs, software APIs)? How were these mechanisms or procedures validated?**

The pollutant readings are collected with the *DALTON* sensing device. Please refer to Appendix A.4 to know how the sensors are validated during data collection. Moreover, a speech-to-text Android application named *vocalAnnot* is used to record annotations from the participants.

**If the dataset is a sample from a larger set, what was the sampling strategy (e.g., deterministic, probabilistic with specific sampling probabilities)?**

The sampling strategy is deterministic.

**Who was involved in the data collection process (e.g., students, crowdworkers, contractors), and how were they compensated (e.g., how much were crowdworkers paid)?**

The occupants of the measurement sites actively participated in the data collection process. The participants include students, university staff, professors, homemakers, canteen owners, etc. The participants received $50 per week as compensation during the field study.

**Over what timeframe was the data collected? Does this timeframe match the creation timeframe of the data associated with the instances (e.g., recent crawl of old news articles)? If not, please describe the timeframe in which the data associated with the instances was created.**

The dataset was collected in 2023-2024. This timeframe matches the creation timeframe of the data.

**Were any ethical review processes conducted (e.g., by an institutional review board)?**

Yes, the ethical review committee of the Indian Institute of Technology Kharagpur has approved the field study (Order No: IIT/SRIC/DEAN/2023, dated 31/07/2023).

## A.4 Correctness

**What exact sensor makes did the authors use for each sensor in the assembled sensing setup?**

We have utilized the particulate matter sensor [53] from DFRobot that uses laser scattering to measure $PM_1$, $PM_{2.5}$, $PM_{10}$ concentration per unit of air volume. The sensor also measures Temperature and Humidity. Moreover, we have utilized the multi-channel gas sensor [54] from Seeed Studio that incorporates four micro-electromechanical systems (MEMS) sensors from Winsen Electronics to measure $NO_2$, Ethanol, VOC, CO. Lastly, to sense $CO_2$, we have used the MH-Z16 Intelligent Infrared Gas sensor [55] from Winsen Electronics. The system specifications and operational details of the sensors are shown in Table 3.

**How did the authors calibrate each sensor?**

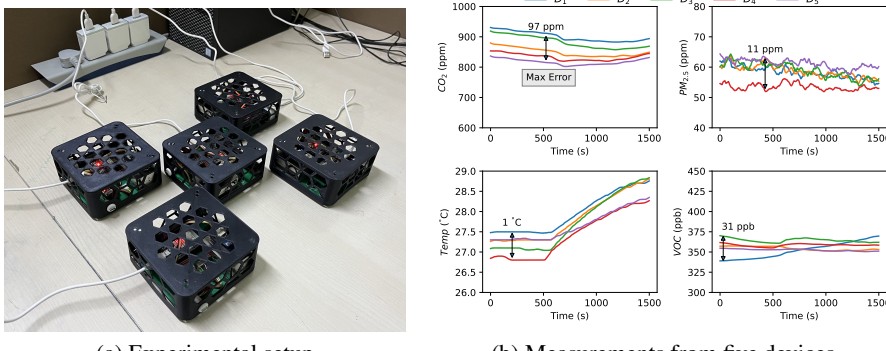

(a) Experimental setup      (b) Measurements from five devices

Figure 7: Readings from five colocated sensing devices indicate the variability across sensors made by the same vendor. The maximum measurement error between two devices is within the error margin, as reported by the vendor.

Although the sensors are factory-calibrated, we have explicitly calibrated each sensor to ensure the correctness of the measurements. We have calibrated the $PM_{2.5}$, Temperature, and Relative Humidity sensors using a reference Airthings device [48]. For the $CO_2$ readings, we have calibrated the MH-Z16 sensor to Zero point (400 ppm) and SPAN point (2000 ppm) as an initial step before the deployment. Further, we have turned on the self-calibration mode of the sensor so that it can judge the zero point intelligently and do the calibration automatically every 24 hours. Moreover, $NO_2$, $C_2H_5OH$, VOC, and CO are one-point calibrated before deployment and periodically cross-checked with a calibrated device during the data collection period.

**How did the authors verify acceptable variability across sensors made by the same vendor?**

The Figure 7a shows five sensing devices colocated in an indoor environment. The measurements shown in Figure 7b validate acceptable (within the reported error margin as shown in Table 3) variability across sensors made by the same vendor.

**Some sensors are very poor at high pollution levels. How did the authors handle this?**

We have used research-grade sensors as listed in Table 3. Indeed, the error margins of those sensors are proportional to the pollution concentration. For instance, the MH-Z16 $CO_2$ sensor is accurate with an error of ±100 ppm+6%value. In a typical household, $CO_2$ concentration goes up to a maximum of 2000-3000 ppm, resulting ±220-280 ppm measurement error. Similarly, the particulate matter has ±10-50 $\mu g/m^3$ measurement error at maximum. In typical households, the sensors are rarely exposed to extremely high pollution levels (unlike industry-level pollution), at which the measurements are significantly erroneous and unusable. Thus, we do not explicitly handle sporadic cases with extremely high pollution levels.

### A.5 Preprocessing/cleaning/labeling

**Was any preprocessing/cleaning/labeling of the data done (e.g., discretization or bucketing, tokenization, part-of-speech tagging, SIFT feature extraction, removal of instances, processing of missing values)? If so, please provide a description.**

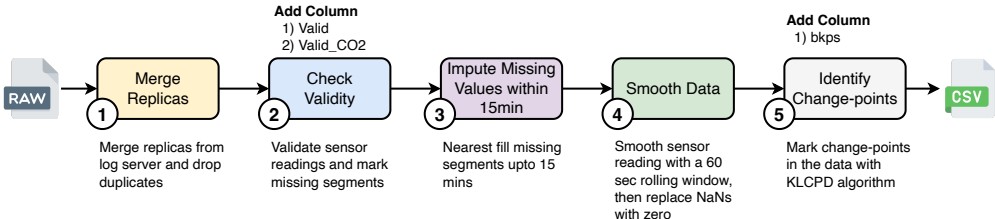

Figure 8: Data preprocessing pipeline.

The dataset is cleaned and organized with the processing pipeline shown in Figure 8. First, the replicas from the log servers are merged, and any duplicates are dropped. Second, the sensor readings are validated based on the reported range of each sensor while marking any missing data segments. Next, missing values are imputed with the nearest fill method if the segment is less than 15 mins. Further, the data is smoothed with a 60-second rolling window to reduce noise before imputing remaining missing values with zero. Finally, the change-points in the sensor readings are marked to denote potential pollution events in the indoor space. After preprocessing, three new columns are computed from the sensor readings, as shown in the figure. The utility of the derived columns is as follows:

- `Valid`: A binary (1/0) column that represents whether all the pollutant readings are within range of the sensors and no sensor is faulty.

- `Valid_CO2`: A binary (1/0) column that represents whether the $CO_2$ sensor is working properly, as it frequently gets impacted due to electrical surges in the indoor sites.

- `bkps`: A binary column (1/0) that marks change-points in the data. The change-points (also known as breakpoints) are computed with the Kernel change point detection (KLCPD) algorithm from the ruptures Python package.

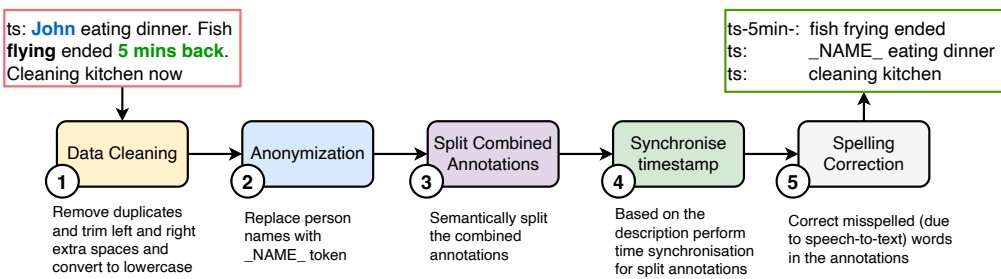

Figure 9: Annotation processing pipeline.

We processed the raw annotations in five steps as shown in Figure 9. First, duplicates are removed, extra spaces are trimmed, and text is converted to lowercase. Second, we replace person names with a _NAME_ token to protect identities. Next, combined annotations are segregated into semantically valid segments as shown in the example in Figure 9. As a fourth step, we perform time synchronization for the split annotations based on their description. Finally, we correct any misspelled words resulting from incorrect speech-to-text conversion (e.g., frying can be annotated as flying or crying) in the annotation Android app, ensuring annotation accuracy and usability of the dataset.

**Was the "raw" data saved in addition to the preprocessed/cleaned/labeled data (e.g., to support unanticipated future uses)? If so, please provide a link or other access point to the "raw" data.**

The raw data files can be downloaded data and are available in a separate Onedrive link (`https://iitkgpacin-my.sharepoint.com/:u:/g/personal/pkarmakar_kgpian_iitkgp_ac_in/EUJjN1c_gU9Jjh2Rj7ghDx8BZOQWS42mP7gHXU8OlHlmjg?e=nzshah`).

**Is the software that was used to preprocess/clean/label the data available? If so, please provide a link or other access point.**

The preprocessing scripts are available in the GitHub repository of the dataset.

## A.6 Uses

**Has the dataset been used for any tasks already? If so, please provide a description.**

The dataset has been used to analyze the spread and accumulation patterns of pollutants in different floor plans and room structures.

**Is there a repository that links to any or all papers or systems that use the dataset? If so, please provide a link or other access point.** No, but we may create one in the future.

**What (other) tasks could the dataset be used for?**

In general, the dataset can be used in the following applications:

- *Pollution Source Identification and Activity Monitoring:* We observe that various pollution sources (i.e., kitchen, disinfectant, etc.) and occupant's activities generate specific pollution patterns based on the activity and how it is performed. The dataset records many such instances, which can be used to learn these unique relationships and develop models for source detection and activity classification.

- *Analysis of Spreading and Accumulation Patterns in Different Floor Plans:* The dataset can be used to analyze the spreading, accumulation, and trapping behavior of indoor pollutants in different indoor floor plans and room structures.

- *Healthy Home Characterization and Improving Designs of Modern Indoors:* The dataset can be used to identify contributory features and design choices of a household that help cope with pollution accumulation and spread, characterizing the healthiness of the household. Further, modern floor plans and room designs can be improved.

- *Smart Device Control (i.e., AC, Exhaust, Air Purifier, etc.):* The dataset can be used to design intelligent control policies to modulate indoor ventilation through precise actuation of exhaust fans, air conditioners, and air purifiers to improve indoor air.

**Is there anything about the composition of the dataset or the way it was collected and preprocessed/cleaned/labeled that might impact future uses?**

We believe that our dataset will not encounter a usage limit.

**Are there tasks for which the dataset should not be used? If so, please provide a description.**

No, users could use our dataset in any task as long as it does not violate laws.

## A.7   Distribution

**Will the dataset be distributed to third parties outside of the entity (e.g., company, institution, organization) on behalf of which the dataset was created? If so, please provide a description.**

No, it will always be held on GitHub.

**How will the dataset will be distributed (e.g., tarball on website, API, GitHub)? Does the dataset have a digital object identifier (DOI)?**

The preprocessing scripts and the dataset are open-sourced and available at `https://github.com/prasenjit52282/dalton-dataset`. Currently, the dataset does not have a digital object identifier.

**When will the dataset be distributed?**

On June 14, 2024.

**Will the dataset be distributed under a copyright or other intellectual property (IP) license, and/or under applicable terms of use (ToU)? If so, please describe this license and/or ToU, and provide a link or other access point to.**

The dataset is released with GNU Affero General Public License: `https://www.gnu.org/licenses/agpl-3.0`. It is free to download for non-commercial research purposes.

**Have any third parties imposed IP-based or other restrictions on the data associated with the instances? If so, please describe these restrictions and provide a link or other access point to, or otherwise reproduce, any relevant licensing terms, as well as any fees associated with these restrictions.**

No third parties are involved in collecting this dataset. The dataset is free for non-commercial usage.

**Do any export controls or other regulatory restrictions apply to the dataset or to individual instances? If so, please describe these restrictions, and provide a link or other access point to, or otherwise reproduce, any supporting documentation.**

No

### A.8 Maintenance

**Who will be supporting/hosting/maintaining the dataset?**

The authors of the paper.

**How can the owner/curator/manager of the dataset be contacted (e.g., email address)?**

Please contact this email address: prasenjitkarmakar52282@gmail.com

**Is there an erratum? If so, please provide a link or other access point.**

Users can use GitHub to report issues or bugs.

**Will the dataset be updated (e.g., to correct labeling errors, add new instances, delete instances)? If so, please describe how often, by whom, and how updates will be communicated to dataset consumers (e.g., mailing list, GitHub)?**

Yes, the authors will actively update the code and data on GitHub.

**If the dataset relates to people, are there applicable limits on the retention of the data associated with the instances (e.g., were the individuals in question told that their data would be retained for a fixed period of time and then deleted)? If so, please describe these limits and explain how they will be enforced.**

All participants signed forms consenting to the use of collected pollutant measurements and activity labels for open-sourced, non-commercial research purposes. Therefore, retention limits are not applicable.

**Will older versions of the dataset continue to be supported/hosted/maintained? If so, please describe how. If not, please describe how its obsolescence will be communicated to dataset consumers.**

Yes, we will provide the information on GitHub.

**If others want to extend/augment/build on/contribute to the dataset, is there a mechanism for them to do so? If so, please provide a description. Will these contributions be validated/verified? If so, please describe how. If not, why not? Is there a process for communicating/distributing these contributions to dataset consumers? If so, please provide a description.**

Yes, we welcome users to submit pull requests on GitHub, and we will actively validate the requests.

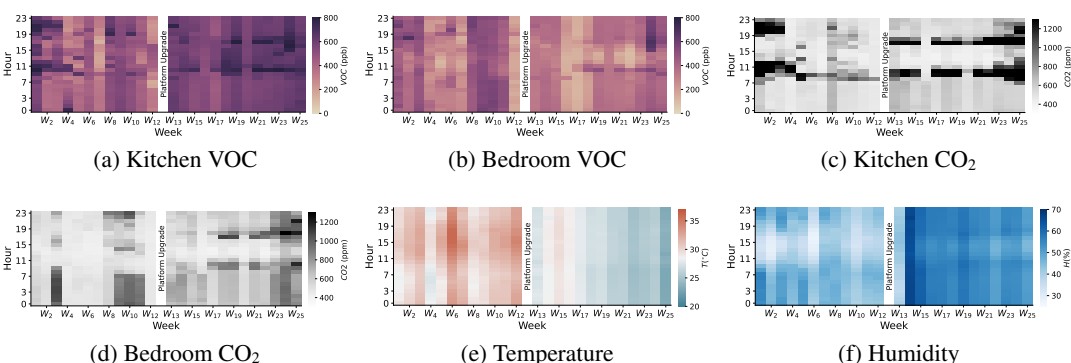

(a) Kitchen VOC     (b) Bedroom VOC     (c) Kitchen CO$_2$

(d) Bedroom CO$_2$     (e) Temperature     (f) Humidity

Figure 10: Daily indoor trends by week and month. We observe higher concentrations of VOC and CO$_2$ during the winter months. The figure is reproduced here and depicted in work [50].

## B  More Dataset Information

Table 6 shows the diversity and scale of the dataset. After the summer season, we upgraded the platform to include remote management features, resuming data collection in the winter.

Figure 10a and Figure 10b show the maximum hourly VOC exposure in the kitchen and bedroom, respectively, revealing a similar pattern that indicates pollutants from the kitchen often spread to

the bedrooms. During the summer (weeks W1 to W12), there is a steady rise in temperature over the weeks (Figure 10e). The overall humidity also increases from week W7 (Figure 10f). High temperatures and humidity cause food items and fruits to degrade quickly, releasing excessive VOCs, which leads to a rise in VOC levels in both kitchens and bedrooms from week W8 onwards.

The highest $CO_2$ peak is observed in the kitchen during the first month when temperatures are relatively comfortable (Figure 10c). People are more sensitive to temperature changes; thus, kitchen exhaust fans are often turned off at comfortable temperatures, resulting in poor ventilation and higher $CO_2$ levels. $CO_2$ peaks decrease as temperatures rise over the months due to the more frequent use of exhaust fans, providing necessary ventilation.

During winter (weeks W13 to W25), the environment becomes more humid, and temperatures steadily decrease. Consequently, occupants tend to keep windows closed to maintain indoor temperatures above 20°C. Kitchen exhausts are also less used due to the low temperatures, and the heat generated during cooking helps improve thermal comfort. As a result, pollution levels increase significantly across the indoor space, with the kitchen becoming the most contaminated area. Pollutants like VOCs and $CO_2$ spread to the bedrooms, and from week W15 onwards, we observe a correlation between $CO_2$ levels in the kitchen and bedrooms.

## B.1 Missing Data

Figure 11 illustrates the distribution of missing data percentage in each file within the dataset. We observe that missing data has an 11.82% mean with a standard deviation of 26.4%. However, the median and mode of missing data distribution are 0.0%, indicating high skewness of the distribution and assuring that most of the files have minimal missing data samples, as shown with a higher density of scatter points on the left side of the figure. The maximum missing data in the files is 97.91%. The box-plot reveals that most files have missing data percentages within the 0-10% range, implying the dataset's quality.

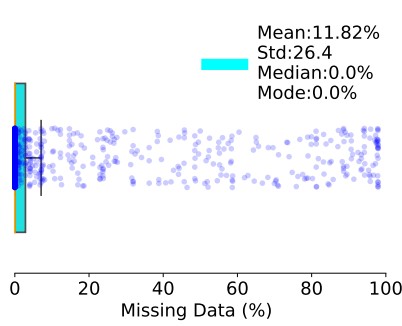

Figure 11: Missing data in the dataset.

## B.2 Application of the Dataset

### B.2.1 Analysis: Typical Pollution Sources & Spread of Pollutants

The dataset provides unique opportunities to analyze the spread of pollutants from typical sources like cooking, indoor gathering, eating, etc., due to multi-device deployment and the contextual activity annotations provided by the participants in the sites. Our prior study [50] explores such aspects of the dataset and shows the spreading, long-term lingering, and trapping of pollutants in various floor plans, room structures, and airflow patterns. Please refer to the study for more details.

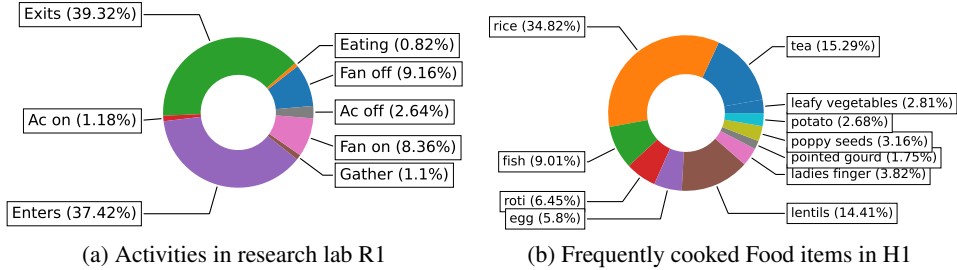

(a) Activities in research lab R1    (b) Frequently cooked Food items in H1

Figure 12: Applications of the dataset and collected data from two sites R1, H1 - (a) detection of indoor activities, (b) Identification of cooked food items from pollution patterns.

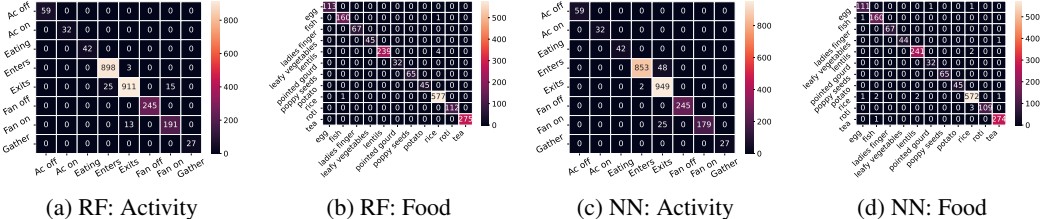

| (a) RF: Activity | (b) RF: Food | (c) NN: Activity | (d) NN: Food |

Figure 13: Confusion Matrix of indoor activity and food item classification with (a)(b) Random Forest with max estimators 50 and max depth 10, (c)(d) Neural Network with three 64-neuron hidden layer.

### B.2.2 ML Application: Identification of Indoor Activities

The dataset can be utilized to understand fluctuations in indoor pollutants (i.e., carbon dioxide, volatile organic compounds, particulate matter) and environmental parameters (i.e., temperature, humidity) that are influenced by indoor activities. We have considered eight indoor activities in a research lab (R1) that can be grouped into three categories based on monitoring motive: (i) Engagement and occupancy (i.e., enter, exit), (ii) Occupant behavior (i.e., fan on/off, AC on/off), (iii) Prohibited practices (i.e., gathering, eating). The class distribution is shown in Figure 12a.

Table 4 summarizes the detailed evaluation of 70-30 random split and 5-fold cross-validation experiments across seven machine learning models with varying parameters. As shown in Figure 13a and Figure 13c, we observe that random forest and neural network misclassify when someone enters or exits the lab. Moreover, exiting and turning on the fan is also confusing for the models. The degree of such confusion increases as we go for simpler models as per Table 4. As we evaluate for even simpler models like logistic regression and naive bayes, the overall performance significantly degrades to 76.4% and 39.9% F1-score, respectively. The primary reason is interleaved activities performed by the lab members, where the pollution signatures get convoluted due to multiple factors. For instance, a member can enter the lab and turn on the fan, resulting in a mixed influence on the pollution data. Moreover, the lab protocol also plays a role by insisting that the members turn on the fans when they leave the lab. Our benchmarking found that the best-performing model, random forest, shows 97.7% testing F1-score. Please refer to our work [51] for more details.

### B.2.3 ML Application: Detection of Cooked Food Items for Automated Food Journalling

The dataset also shows potential applications for food item classification from the pollution signatures captured in the kitchen of household H1. We have considered eleven frequently cooked foods (i.e., ladies finger, lentils, egg, fish, poppy seeds, potato, pointed gourd, rice, roti, leafy vegetables, tea). The class distribution is shown in Figure 12b. With similar features computed in our work [51], we reported our benchmarking on seven machine learning algorithms in Table 5 for classifying the food items. As per the table, random forest and neural network perform best with 98.3% and 99.5% testing F1-score in 70-30 split experiments (see the confusion matrix in Figure 13b and Figure 13d). However, in 5-fold cross-validation, the performance drops due to class imbalance and fewer cooking instances for only using H1 (as cooking generally occurs only twice a day).

We performed the benchmarking experiments on a Workstation that is equipped with an Intel i9-12900K 24-core CPU, 64GB primary memory, and NVIDIA RTX A4500 20GB GPU, running Ubuntu 22.04.3 LTS and Python 3.11.5. As discussed in this paper, our benchmarking indicates the dataset's utility for several downstream ML applications using pollution data from multiple sensors and available activity annotations.

### B.3 Additional Insights

The Figure 14a and Figure 14b present the spatiotemporal spread of VOC and $CO_2$ in H1 during and after cooking activity in Kitchen (A). Cooking starts at $t$ time and gets over by $t+11$ mins. We

Table 4: Performance of the ML models in 70-30 random split and 5-fold cross-validation experiments in activity detection.

| Model | Parameters | 70-30 Random Split | | | | | | 5-Fold Cross-validation | | | |
|---|---|---|---|---|---|---|---|---|---|---|---|
| | | Training (Weighted) | | | Testing (Weighted) | | | Accuracy (Mean) | | Accuracy (Std) | |
| | | F1-score | Precision | Recall | F1-score | Precision | Recall | Train | Test | Train | Test |
| SVM | Linear kernel | 0.831 | 0.837 | 0.833 | 0.817 | 0.827 | 0.821 | 0.495 | 0.497 | 0.0156 | 0.0176 |
| | Polynomial kernel | **0.892** | **0.894** | **0.892** | **0.879** | **0.881** | **0.879** | **0.543** | **0.542** | **0.0041** | **0.0075** |
| | RBF kernel | 0.815 | 0.836 | 0.82 | 0.798 | 0.82 | 0.804 | 0.53 | 0.53 | 0.0036 | 0.01 |
| Naive Bayes | Gaussian | 0.403 | 0.727 | 0.399 | 0.399 | 0.717 | 0.39 | 0.424 | 0.419 | 0.0105 | 0.0047 |
| Decision Tree | Max depth10 | 0.949 | 0.95 | 0.949 | 0.924 | 0.924 | 0.924 | 0.967 | 0.951 | 0.0025 | 0.0051 |
| | Max depth 20 | 0.992 | 0.992 | 0.992 | 0.975 | 0.975 | 0.976 | 0.992 | 0.974 | 0.0008 | 0.0039 |
| | Max depth 30 | **0.992** | **0.992** | **0.992** | **0.976** | **0.976** | **0.976** | **0.992** | **0.975** | **0.0007** | **0.0039** |
| | Max depth 40 | 0.992 | 0.992 | 0.992 | 0.976 | 0.976 | 0.976 | 0.992 | 0.975 | 0.0007 | 0.0039 |
| k-Nearest Neighbour | Neighbour 10 | **0.981** | **0.981** | **0.981** | **0.975** | **0.975** | **0.975** | **0.985** | **0.979** | **0.0008** | **0.0022** |
| | Neighbour 20 | 0.972 | 0.972 | 0.972 | 0.967 | 0.967 | 0.967 | 0.983 | 0.979 | 0.0002 | 0.0024 |
| | Neighbour 30 | 0.959 | 0.96 | 0.96 | 0.956 | 0.956 | 0.957 | 0.979 | 0.976 | 0.0015 | 0.0044 |
| | Neighbour 40 | 0.946 | 0.946 | 0.947 | 0.947 | 0.947 | 0.947 | 0.975 | 0.972 | 0.0021 | 0.0036 |
| Logistic Regression | – | 0.791 | 0.801 | 0.796 | 0.764 | 0.777 | 0.77 | 0.581 | 0.577 | 0.0088 | 0.0145 |
| Random Forest | Max estimator 30 Max depth 10 | 0.988 | 0.988 | 0.988 | 0.979 | 0.979 | 0.979 | 0.989 | 0.977 | 0.0005 | 0.0021 |
| | Max estimator 50 Max depth 10 | **0.988** | **0.988** | **0.988** | **0.977** | **0.977** | **0.977** | **0.989** | **0.979** | **0.0006** | **0.0049** |
| | Max estimator 100 Max depth 10 | 0.99 | 0.99 | 0.99 | 0.979 | 0.979 | 0.979 | 0.989 | 0.977 | 0.0012 | 0.0037 |
| Neural Network | Hidden [64, 64] | 0.981 | 0.981 | 0.981 | 0.973 | 0.973 | 0.973 | 0.925 | 0.92 | 0.0229 | 0.0165 |
| | Hidden [64, 64, 64] | **0.982** | **0.983** | **0.982** | **0.978** | **0.979** | **0.978** | **0.947** | **0.943** | **0.0104** | **0.0135** |
| | Hidden [128, 128] | 0.981 | 0.982 | 0.981 | 0.979 | 0.979 | 0.979 | 0.912 | 0.91 | 0.0116 | 0.0114 |
| | Hidden [128, 128, 128] | 0.982 | 0.982 | 0.982 | 0.978 | 0.978 | 0.978 | 0.95 | 0.946 | 0.0174 | 0.0203 |

Table 5: Performance of the ML models in 70-30 random split and 5-fold cross-validation experiments in food item detection.

| model | Parameters | 70-30 Random Split | | | | | | 5-Fold Cross-validation | | | |
|---|---|---|---|---|---|---|---|---|---|---|---|
| | | Training (Weighted) | | | Testing (Weighted) | | | Accuracy (Mean) | | Accuracy (Std) | |
| | | F1-score | Precision | Recall | F1-score | Precision | Recall | Train | Test | Train | Test |
| SVM | Linear kernel | 0.741 | 0.781 | 0.754 | 0.734 | 0.781 | 0.747 | 0.681 | 0.33 | 0.064 | 0.1041 |
| | Polynomial kernel | **0.865** | **0.895** | **0.868** | **0.866** | **0.897** | **0.87** | **0.601** | **0.426** | **0.0133** | **0.0592** |
| | RBF kernel | 0.838 | 0.873 | 0.842 | 0.837 | 0.871 | 0.841 | 0.625 | 0.419 | 0.0134 | 0.0809 |
| Naive Bayes | Gaussian | 0.334 | 0.526 | 0.318 | 0.334 | 0.516 | 0.317 | 0.363 | 0.223 | 0.0206 | 0.0242 |
| Decision Tree | Max depth10 | 0.974 | 0.975 | 0.974 | 0.961 | 0.962 | 0.961 | 0.941 | 0.387 | 0.0328 | 0.064 |
| | Max depth20 | **0.999** | **0.999** | **0.999** | **0.984** | **0.984** | **0.984** | **0.999** | **0.374** | **0.0003** | **0.0634** |
| | Max depth30 | 0.999 | 0.999 | 0.999 | 0.984 | 0.984 | 0.984 | 0.999 | 0.374 | 0.0003 | 0.0634 |
| | Max depth40 | 0.999 | 0.999 | 0.999 | 0.984 | 0.984 | 0.984 | 0.999 | 0.374 | 0.0003 | 0.0634 |
| k-Nearest Neighbour | Neighbour 10 | **0.991** | **0.991** | **0.991** | **0.982** | **0.982** | **0.982** | **0.965** | **0.407** | **0.0047** | **0.0885** |
| | Neighbour 20 | 0.963 | 0.964 | 0.963 | 0.948 | 0.949 | 0.948 | 0.921 | 0.388 | 0.008 | 0.0836 |
| | Neighbour 30 | 0.921 | 0.922 | 0.921 | 0.912 | 0.914 | 0.912 | 0.864 | 0.386 | 0.0112 | 0.0732 |
| | Neighbour 40 | 0.847 | 0.851 | 0.849 | 0.849 | 0.857 | 0.852 | 0.823 | 0.368 | 0.0172 | 0.0724 |
| Logistic Regression | – | 0.659 | 0.695 | 0.686 | 0.652 | 0.692 | 0.679 | 0.603 | 0.415 | 0.0231 | 0.0674 |
| Random Forest | Max estimator 30 Max depth 10 | 0.988 | 0.988 | 0.988 | 0.978 | 0.98 | 0.979 | 0.995 | 0.521 | 0.0033 | 0.1361 |
| | Max estimator 50 Max depth 10 | **0.991** | **0.991** | **0.991** | **0.982** | **0.983** | **0.982** | **0.993** | **0.532** | **0.0056** | **0.1298** |
| | Max estimator 100 Max depth 10 | 0.997 | 0.997 | 0.997 | 0.994 | 0.994 | 0.994 | 0.996 | 0.524 | 0.0031 | 0.1234 |
| Neural Network | Hidden [64, 64] | 0.99 | 0.99 | 0.989 | 0.983 | 0.983 | 0.983 | 0.799 | 0.368 | 0.057 | 0.0539 |
| | Hidden [64, 64, 64] | 0.998 | 0.998 | 0.998 | 0.996 | 0.996 | 0.996 | 0.913 | 0.42 | 0.0522 | 0.0928 |
| | Hidden [128, 128] | 0.999 | 0.999 | 0.999 | 0.994 | 0.994 | 0.994 | 0.892 | 0.434 | 0.0385 | 0.1216 |
| | Hidden [128, 128, 128] | **0.997** | **0.997** | **0.997** | **0.995** | **0.995** | **0.995** | **0.965** | **0.443** | **0.0226** | **0.1446** |

observe that VOC is hard to ventilate compared to $CO_2$. VOC spreads even after 6 minutes (i.e., $t$+17 mins), whereas $CO_2$ depletes to normal levels. In Figure 14a (subfigure $t$+48 mins), we observe trapping of VOC in the bedroom (E).

The Figure 15a and Figure 15b shows the spread of VOC and $CO_2$ in H4 during and after cooking activity in Kitchen (A). Cooking ended at $t$+33 mins. Unlike H1, we observe increased levels of VOC and $CO_2$ even after 14 mins (i.e., Figure 15, subfigures $t$+48 mins). Moreover, long-term lingering and trapping of the VOC can be observed in Figure 15a (subfigure $t$+90 mins) due to the congested room structure of H4.

Table 6: Summarization of the overall deployment, user participation, and data collection scale across 30 diverse sites spread across four geographic regions in India.

| Site ID (Lat,Long) | Devices | #Devices | Site Area (Sqft) | Floor Plan | #F/#M | Duration (Hrs) | #Samples | Activity Annot. (Hrs) | Occupants |
|---|---|---|---|---|---|---|---|---|---|
| H1 (23.426773806721958, 87.28093138944492) | 41 Kitchen
42 Bedroom beside
43 Dining left
44 Dining right
45 Parent room | 5 | 1100 | ✓ | 1/1 | 772 | 11402870 | 768 | **P1**
**P2** |
| H2 (22.311780284326424, 87.30473175163596) | 13 Ground floor bedroom
17 Blue room bedside
16 Blue room north
14 Bedroom router
12 Kitchen window
15 Orange room south
11 Front of TV | 7 | 1100 | ✓ | 2/2 | 469 | 8333689 | 336 | **P3**
P4
P5
P6 |
| H3 (22.30909193614537, 87.30266133055456) | 62 Kitchen RO
63 Dining
61 Bed room desk | 3 | 1000 | ✓ | 1/1 | 463 | 4041058 | 463 | **P7**
**P8** |
| H4 (23.551011187302482, 87.29059861761522) | 13 Kitchen
11 Bedroom
12 Bedroom Ma
15 Dining room
14 TV room | 5 | 1200 | ✓ | 1/1 | 2635 | 24021924 | – | P9
P10 |
| H5 (23.551200930425804, 87.28961511893861) | 22 TV room
21 Dining room | 2 | 1200 | ✓ | 1/1 | 2634 | 7395189 | – | P11
P12 |
| H6 (22.313294885316072, 87.3082087768644) | 114 Space1
113 Room3
111 Room1
112 Room2
115 Space2 | 5 | 400 | ✓ | 1/1 | 218 | 3188644 | 96 | **P13**
P14 |
| H7 (22.31171268768513, 87.3045281731776) | 71 Kitchen door
72 Dining fridge | 2 | 400 | – | 1/1 | 366 | 2306882 | 366 | **P15**
**P16** |
| H8 (23.426867453740456, 87.28087091558989) | 84 Dinning fridge
81 Kitchen
83 Dinning
82 Bedoom
85 bedroom2 | 5 | 1100 | – | 2/1 | 570 | 8676832 | 480 | **P1**
P17
P18 |
| H9 (22.665393416524967, 88.3758336145021) | 98 Kitchen
99 Bedroom | 2 | 300 | – | 1/1 | 768 | 3894082 | 480 | **P19**
P20 |
| H10 (22.567985465871004, 88.36845057670452) | 104 Sealdah2
103 Sealdah1 | 2 | 600 | – | 2/2 | 25 | 70554 | – | P21 P22
P23 P24 |
| H11 (22.66496174846917, 88.37682253157242) | 107 Dining
106 Kitchen | 2 | 600 | – | 1/2 | 86 | 60098 | – | P25 P26
P27 |
| H12 (21.900240213668614, 87.53801060138943) | 94 Bedroom
93 Kitchen | 2 | 216 | – | 1/1 | 178 | 1054696 | 144 | **P19**
P20 |
| H13 (22.652632984217956, 88.41135038509114) | 96 Bedroom
95 Kitchen | 2 | 216 | – | 1/1 | 127 | 269824 | 24 | **P19**
P20 |
| A1 (22.316371948187594, 87.30576795184774) | 101 Study Desk | 1 | 150 | – | 1/0 | 146 | 226888 | 48 | **P28** |
| A2 ((22.30971013354728, 87.30784886676756)) | 105 Study Desk | 1 | 150 | – | 0/1 | 289 | 193557 | – | P29 |
| A3 (22.30937218266447, 87.31064479668473) | 109 Study Desk | 1 | 180 | – | 0/1 | 344 | 1098827 | 96 | **P30** |
| A4 (22.318929727398313, 87.30650918833601) | 120 Study Desk | 1 | 150 | – | 1/0 | 125 | 384975 | – | P31 |
| A5 22.318929727398313, 87.30650918833601 | 121 Study Desk | 1 | 150 | – | 1/0 | 1 | 77 | 1 | **P32** |
| A6 (22.31648291562444, 87.29538496948412) | 122 Study Desk | 1 | 100 | – | 0/1 | 51 | 154398 | 51 | **P33** |
| A7 (22.30971013354728, 87.30784886676756) | 113 On bed | 1 | 150 | – | 0/1 | 55 | 54741 | 55 | **P34** |
| A8 (22.30971013354728, 87.30784886676756) | 151 Study Desk | 1 | 150 | – | 0/1 | 60 | 189141 | – | P35 |
| R1 (22.3179564762161, 87.30920753796912) | 54 Desk4
51 Desk1
53 Desk3
52 Desk2 | 4 | 522 | ✓ | 1/6 | 834 | 6203065 | 528 | **P36 P37**
**P38 P39**
**P40 P41**
**P42** |
| R2 (22.3179564762161, 87.30920753796912) | 102 Desk Lab 319 | 1 | 320 | ✓ | 2/2 | 367 | 1161570 | 72 | **P43** |
| R3 (22.3179564762161, 87.30920753796912) | 108 Left Desk | 1 | 616 | ✓ | 0/1 | 243 | 750745 | 72 | **P44** |
| R4 (22.3179564762161, 87.30920753796912) | 21 Desk1
24 Desk4
22 Desk2
23 Desk3 | 4 | 522 | ✓ | – | 371 | 387195 | – | – |
| R5 (22.3179564762161, 87.30920753796912) | 73 Desk3
72 Desk2
71 Desk1 | 3 | 600 | ✓ | – | 179 | 1583750 | – | – |
| F1 (22.30936287704651, 87.31055134200086) | 23 Kitchen | 1 | 150 | ✓ | 2/0 | 450 | 631193 | – | P46 |
| F2 (22.32048015405153, 87.30800205041001) | 24 Kitchen | 1 | 150 | ✓ | – | 450 | 631193 | – | – |
| C1 (22.3179564762161, 87.30920753796912) | 25 Teacher Desk | 1 | 500 | – | – | 333 | 590272 | – | – |
| C2 (22.3179564762161, 87.30920753796912) | 26 Teacher Desk | 1 | 500 | – | – | 53 | 158256 | – | – |

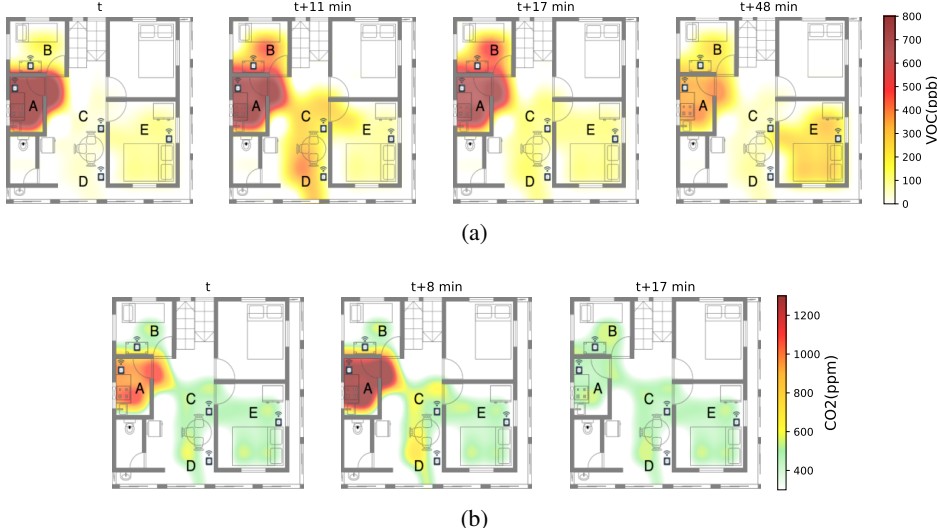

Figure 14: Spatiotemporal spread of – (a) VOC, and (b) $CO_2$ from the kitchen in Household H1. The figure is reproduced to provide insights on the dataset and depicted in work [50].

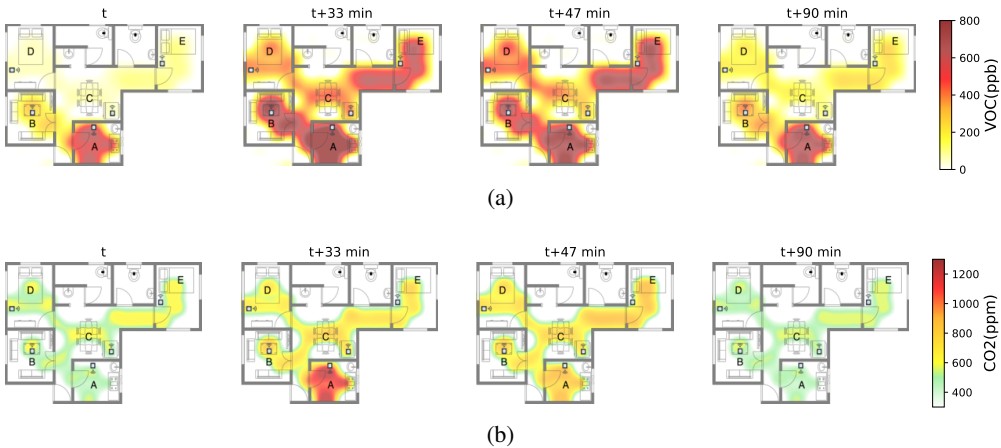

Figure 15: Spatiotemporal spread of – (a) VOC, and (b) $CO_2$ from the kitchen in Household H4. The figure is reproduced to provide insights on the dataset and depicted in work [50].

