# OpenReview forum: "Indoor Air Quality Dataset with Activities of Daily Living in Low to Middle-income Communities"
_NeurIPS.cc/2024/Datasets_and_Benchmarks_Track — NeurIPS 2024 Track Datasets and Benchmarks Poster_

### Official Review · Reviewer_97sg · 2024-07-20

**Rating:** 7
**Confidence:** 4

**Review:**

Overall, this is an important contribution as it seems like a first large dataset from the region.

Clarity
The paper exhibits good clarity. The documentation on Github is clear. The dataset seems easily accessible.

Significance.
The dataset is valuable. However, perhaps due to a lack of space, the presented analysis seems to mostly concern anecdotal evidences, and it is not clear how the mentioned potential applications would be realised.

**Strengths:**

- The paper collects data over a long duration across different kinds of households and units.
- The paper also presents user annotations to help contextualise the collected data
- The anecdotal evidences are useful to help understand the effects of different controls
- Using multiple sensors and thus looking at correlation and causations would be useful
- While there are millions of air quality sensors deployed across the world, there is in general a dearth of research grade datasets. This dataset can overcome that.

**Additional Feedback:**

I have mentioned my feedback in earlier sections.

**Clarity:**

Overall, the paper is well written. It flows well and covers the main sections in sufficient detail.

**Correctness:**

The collected data setting looks fine. But, I am not sure and as the authors point out the annotations by humans can be wrong. Perhaps, there can be more automated ways of collecting the additional context collected by the authors. For example, using contact sensors to check if an appliance (like the exhaust) is On/Off.

Further, there is no discussion on the correctness of the sensing equipment itself. Is the sensing equipment gold standard? If so, this needs to be clarified. If not, does it need to be calibrated?

**Documentation:**

The documentation seems reasonable.

**Ethics:**

No.

**Limitations:**

The presented limitations do not have any negative social impact in my opinion.

Minor comment -- sometimes people faces are blurred in research articles.

**Opportunities For Improvement:**

- The authors mention missing data due to outages, etc. It would be useful to quantify such statistics.
- One of my main concerns is that the paper while present potential applications, does not demonstrate those.
    - I mean the utility to the ML community is not clear.  Perhaps this could be fixed with writing or some minimal experiments. Some of the use cases are presented as anecdotal evidences but not really as prediction problems.
    - The utility to the other domains is not clear. Perhaps again this is possible but just a writing issue; or could be solved with some minimal experiments.
         - Some of the related work or enabling work from this dataset could be for example for optimising air, energy (HVAC) and comfort. But, it is not clear if this data (as it is presented) can enable such control.
        - For healthcare work, studies have looked at low-cost sensors and looked at longitudinal studies but measured health parameters. Would your dataset enable such studies?

- Are there other large datasets? How does this differ from them?

**Relation To Prior Work:**

I feel overall the paper discusses the prior work well.


However, I feel the authors may consider a table of comparison.

Some more papers I found
1. https://data.mendeley.com/datasets/2r232jpfb2/1
2. https://www.nature.com/articles/s41597-023-02640-y

**Summary And Contributions:**

The paper presents a dataset of indoor air quality collected from four sites in India (LMIC). Overall, the dataset is a useful contribution as there are not many existing datasets of similar diversity and volume. Such a dataset can potentially enable applications in: control, health sensing, floor plan design, etc.

---

> ### Author Rebuttal · Authors · 2024-08-17
>
> Thanks for appreciating our effort. Here are the specific responses to your individual review comments.
>
> 1. **[Missing data statistics due to outage]** We plan to include a Figure 5 in the supplementary material that illustrates the distribution of missing data percentage in each file within the dataset. We observe that missing data has an 11.82% mean with a standard deviation of 26.4%. However, the median and mode of missing data distribution are 0.0%, indicating high skewness of the distribution and assuring that most of the files have minimal missing data samples, as shown with a higher density of scatter points on the left side of the figure. The maximum missing data in the files is 97.91%. In most files, the missing data percentage is within the 0-10% range, indicating that the dataset is of high quality.
>
> 2. **[Demonstration of potential applications]** We plan to demonstrate two possible applications of the dataset in the supplementary material: (i) indoor activity recognition (i.e., enter, exit, fan on, fan off, AC on, AC off, gathering, eating), (ii) food item classification (i.e.,  ladies finger, lentils, egg, fish, poppy seeds, potato, pointed gourd, rice, roti, leafy vegetables, tea) from pollution emission. We have benchmarked seven commonly used ML models. We observe 0.977 and 0.982 testing F1-score with Random Forest for the above two tasks. Please check Figure 7, Tables 2 and 3 in the attachment that we plan to include in the revised supplementary material for the detailed results with various other benchmarking models.
>
> 3. **[Utility to other domains]** We believe the indoor air pollution dataset is an important context information not just for healthy living but also for different context aware smart home applications. This also complements the huge array of activities of daily living (ADL) dataset which mostly defines type and duration of the activity but not the impact of activity on the environment and one’s health. For instance, we plan to include an additional Figure 5a in the primary manuscript (please check the attachment for the figure), which shows the increase in pollutants when the user forgot to turn on the exhaust fan in the kitchen while cooking. Similarly, Figure 5b shows the impact of incense sticks and mosquito repellent on the environment. Therefore, this dataset can be used to motivate in designing and building healthier indoors (i.e, structural engineering and architecture), physical activity coupled with pollution aware recommendation engines (such as nudging to open windows, turn on ventilation or songs related to joy of cooking played in voice assistants), etc.
>
> 4. **[Healthcare studies]** Long-term CO2 exposure causes occupants to experience less concentrations, stress and anxiety [Savulich et al.]. As a result, the dataset can be used to predict such scenarios beforehand to reduce their health impact. Although we do not directly collect any health parameters, as providing each participant with an applicable wearable (such as the Empatica Watch) to access the collected data was not financially feasible. It will be included in our future work discussions in the paper.
>
> [Savulich et al.] Savulich, G., Hezemans, F.H., van Ghesel Grothe, S., Dafflon, J., Schulten, N., Brühl, A.B., Sahakian, B.J. and Robbins, T.W., 2019. Acute anxiety and autonomic arousal induced by CO2 inhalation impairs prefrontal executive functions in healthy humans. Translational Psychiatry, 9(1), p.296.
>
> 5. **[Other large datasets]** Thank you for your suggestion. Nonetheless, we conducted a detailed comparative study in the related work. The [ASHRAE] database is there that comprises all the datasets discussed in the paper. In none of them, daily activities are collected since they require active participation from the occupants for the entire study period. The existing datasets mostly include events from the smart plugs, HVAC, door, and light sensors. Such datasets are relevant for analyzing building energy efficiency and user comfort. We also integrate diverse activities of daily living with a large-scale measurement of the indoor pollutants from multiple sites, enabling analysis over the direct (short-term) or in-direct (long-term) impact of occupant behavior on indoor pollutants.
>
> [ASHRAE] Dong, B., Liu, Y., Mu, W., Jiang, Z., Pandey, P., Hong, T., Olesen, B., Lawrence, T., O’Neil, Z., Andrews, C. and Azar, E., 2022. A global building occupant behavior database. Scientific data, 9(1), p.369.
>
> 6. **[Correctness of the sensing devices]** We have provided calibration details in the supplementary sheet (see Section 1.4 “Correctness”).
>
> 7. **[Other relevant papers suggested by the reviewer]** Thank you for your appreciation and suggestion. However, we have cited (b) in our related work and compared it against the Dalton-dataset. Moreover, we cited a more relevant dataset available in kaggle and mendeley (https://data.mendeley.com/datasets/kn3x9rz3kd/1) in our related work that have activity labels only for normal situation, preparing meals, presence of smoke, and cleaning. In contrast, (a) lacks any activity labels that are crucial to understanding the context of indoor pollution.
> https://data.mendeley.com/datasets/2r232jpfb2/1
> https://www.nature.com/articles/s41597-023-02640-y
>
> Please feel free to let us know if you have any further suggestions, comments or concerns. Thank you again for your time.

---

> > ### Comment · Reviewer_97sg · 2024-08-19
> >
> > Thank you for addressing my comments. I am happy to see the changes and am happy to update my rating by 1 point.

---

> ### Author Rebuttal · Authors · 2024-08-19
>
> Thank you for appreciating the work. We'll make sure to include the details in the paper.

---

### Official Review · Reviewer_2kUv · 2024-07-22
**Important Dataset in the Field**

**Rating:** 7
**Confidence:** 4
**Correctness:** Yes
**Clarity:** Yes

**Review:**

The paper presents a valuable indoor air quality dataset (along with meteorology and other factors) with enough diversity, temporal resolution, and time period. Rarely do we see datasets where authors build their own devices, physically go to the field and deploy the sensors, maintain them, and then release a dataset to the research community. Doing such experiments in developing countries has its own challenges.

The human activity dataset will help other researchers dive deeper and bring out more insights, enabling them to reason about the concentrations of various pollutants. Involving 46 participants for the human activity data was a good addition to the air quality dataset.

Edit1: Reviewer cXtY has raised some valid issues with the data and if they are not addressed during the rebuttal, I may reduce my rating.

**Strengths:**

* Clear flow of the paper. Easy to follow, evaluate, and read.
* High temporal resolution of the data.
* Diversity of the data: i) selection of participants (with different backgrounds); ii) geographical region; ii) types of the sites (e.g. home, canteen etc.); iv) number of sensors per site.
* Insights from the dataset are discussed in detail e.g. split AC v/s window AC case, CO2 levels in the kitchen (importance of exhaust fan)

**Additional Feedback:**

NA

**Documentation:**

Yes

**Ethics:**

The authors involve human participants and pay them appropriate remuneration during the study period. They have also obtained ethical approval from the institute's ethical review committee.

**Limitations:**

Yes, the limitations are discussed and are valid.

**Opportunities For Improvement:**

> The 3D-printed shell features a hollow honeycomb structure that allows unbiased measurement of the pollutants at one sample per second (1 Hz) frequency
* Above line seems to suggest the temporal resolution of the dataset but I'd suggest the authors to explicitly mention the temporal resolution of the dataset. Was it consistent across all the sites (apart from missing data due to power failures)?

Other questions:
* At what rate was the data sent to the cloud (e.g., every 1 minute)? Using which protocol (e.g. MQTT) or private/public APIs?
* What storage service was used to store the dataset on the cloud? Did you use any specific API from platforms like Google Cloud Platform? If yes, specifying them can help other researchers working on similar IoT devices.
* "Dataset Analysis" section is mainly focused on CO2 and touches upon VOC. Can you comment on PM2.5 as well which is generally perceived as a highly unhealthy pollutant? It'd also be interesting to see if other pollutants are discussed with similar insights (perhaps in supplementary if not in the main paper).
* It could be helpful to tap a little more into "Possible Dataset Applications" and concretize it in later sections. For example, an annotated image dataset would be more useful if accompanied by results from a set of classification algorithms to establish the utility of the dataset. The experiments need not be extensive but should be sufficient without loss of generality.

**Relation To Prior Work:**

Yes, the paper discussed the prior work in sufficient detail.

**Summary And Contributions:**

The paper collects and releases a high temporal resolution indoor air quality dataset along with human-in-the-loop speech-to-text activity data from 30 sites over four regions in India for six months. The study sites involve diverse places such as homes, labs, canteen, and studio apartments.

The main contributions of the paper are the following:
* Authors release a unique, diverse and high temporal resolution indoor air pollution data to the research community.
* Authors have induced the diversity in: i) selection of participants (with different backgrounds); ii) geographical region; ii) types of the sites (e.g. home, canteen etc.); iv) number of sensors per site.
* Authors provide insights from the datasets in various scenarios (e.g. effect of split AC v/s window AC).
* Authors have mentioned the exact sensors/hardware used to build their in-house measuring device in the supplementary material.

---

> ### Author Rebuttal · Authors · 2024-08-17
>
> Thanks for appreciating our effort. Here are the specific responses to your individual review comments. We have also addressed the issue as pointed out by Reviewer cXtY.
>
> 1. **[Temporal resolution of the dataset]** The temporal resolution of the raw dataset is 1 sample per second, and is consistent across all the deployment sites. We plan to mention this explicitly in the revised manuscript.
>
> 2. **[Data rate to the cloud and storage service]** We have sent the data to the cloud at a rate of 1 sample per second via MQTT. We have used Microsoft Azure for VMs. We have stored the data in a MongoDB instance.
>
> 3. **[PM2.5 and other pollutants]** PM2.5 is not significant for indoors unless cooking is happening. We plan to highlight a particular scenario in Figure 6 of the primary manuscript (please check the attachment for this figure) where we show how PM2.5 can spread towards interior rooms due to swirling airflow from the ceiling fan. The kitchen emits pollutants from t to t + 11 minutes. Lighter dust particles PM1 and PM2.5 spread more aggressively compared to heavier PM10 particles. We also plan to include additional observations on the spreading patterns of VOC and CO2 in the supplementary material. In addition, we plan to refer to our recent [JCSS] paper for more pollution spread analysis.
>
> [JCSS]
> Karmakar, P., Pradhan, S. and Chakraborty, S., 2024. Exploring Indoor Air Quality Dynamics in Developing Nations: A Perspective from India. ACM Journal on Computing and Sustainable Societies.
>
> 4. **[Possible dataset application]** Thanks for this comment. We plan to include two pilot observations for two possible ML applications of the dataset in the supplementary material: first, indoor activity detection (i.e., enter, exit, fan on, fan off, AC on, AC off, gathering, eating) from pollution signatures (we plan to include an additional Table 2 in the supplementary material, please check the attachment), and second, food item (i.e., ladies finger, lentils, egg, fish, poppy seeds, potato, pointed gourd, rice, roti, leafy vegetables, tea) classification from pollution emission while cooking (we plan to include an additional Table 3 in the supplementary material, please check the attachment). We have benchmarked seven commonly used classification models. We observe 0.977 and 0.982 testing F1-score with Random Forest for the above two tasks, establishing the utility of the dataset. Please check Figure 7, Tables 2 and 3 in the attachment, which we plan to include in the revised supplementary material for the details with other benchmarking models.
>
> Please feel free to let us know if you have any further suggestions, comments or concerns. Thank you again for your time.

---

### Official Review · Reviewer_TNmg · 2024-07-24

**Rating:** 7
**Confidence:** 4
**Clarity:** The paper is well written to a wider …

**Review:**

The paper is well written, the data collection procedure is well motivated and documented. This would be of broader interest to the community.

**Strengths:**

* The study is conducted across 4 regions in India covering different socio-economic backgrounds, and activity types. The participants involved are students, homemakers, canteen staff, and professors - thus ensuring diversity in gender, profession, and activity profiles
* The data is annotated with activity for the duration of the increased/decreased pollution - which is instrumental in source attribution
* Metadata such as square footage of the indoor location, floor plan, and demographic information of the participants are provided to perform fine-grained analysis

**Additional Feedback:**

N/A

**Correctness:**

Benchmarking against pollution prediction and activity recognition models would convince the readers that the data is unique for future analysis.

**Documentation:**

The README file in the URL carefully outlines the features and paths available for the raw and processed data in a usable format.

**Ethics:**

The IRB approval notice should be made public to ensure that the terms of use of the data is clarified in the dataset.

**Limitations:**

A discussion of the privacy preservation steps, and how that could be circumvented needs to be discussed.

**Opportunities For Improvement:**

* In addition to qualitative analyses provided in the paper, the soundness of the result can be improved by benchmarking existing activity recognition or pollution prediction models to better profile the uniqueness of the dataset
* Standardization of the processing of multivariate data such as floor plans, activity should be provided in the manuscript/appendix for better readability.
* A discussion on missingness in the data due to power outages, and calibration of the sensors deployed should be provided to improve the adoption of the dataset for modeling and analysis tasks.

**Relation To Prior Work:**

Related work is well studied and positioned

**Summary And Contributions:**

The paper provides an indoor air quality dataset collected within low-middle income households, canteens, and labs with multiple sensors measuring 8 known pollutant levels throughout the day. Further, the data is annotated with activity gathered through a speech-to-text tool to identify the sources behind the pollutant level changes. Finally, the data is annotated with ventilation types, floor plans, etc, which would aid future work in analyzing and mitigating indoor air pollution.

---

> ### Author Rebuttal · Authors · 2024-08-17
>
> Thanks for appreciating our effort. Here are the specific responses to your individual review comments.
>
> 1. **[Benchmarking the activities]** Thanks a lot for pointing this out. The focus of this paper was a holistic indoor pollution dataset and we have already published some companion works around different possible downstream applications. For example, we use this dataset for lab activity monitoring [MobileHCI], and cooking driven pollution spread analysis [JCSS], etc. Following the suggestions from the reviewer, we have included additional benchmarking results from lab activity monitoring (Figure 7a, 7c and Table 2, please check the attachment) and food item classification (Figure 7b, 7d and Table 3, please check the attachment) in the supplementary material. We also plan to refer to the [JCSS] paper in the revised manuscript for highlighting the pollution spread analysis to demonstrate the uniqueness of the dataset.
>
> [JCSS]
> Karmakar, P., Pradhan, S. and Chakraborty, S., 2024. Exploring Indoor Air Quality Dynamics in Developing Nations: A Perspective from India. ACM Journal on Computing and Sustainable Societies. DOI: https://doi.org/10.1145/3685694
>
> [MobileHCI]
> Prasenjit Karmakar, Swadhin Pradhan, and Sandip Chakraborty. 2024. Exploit-
> ing Air Quality Monitors to Perform Indoor Surveillance: Academic Setting.
> arXiv:2408.05779 [cs.HC] https://arxiv.org/abs/2408.05779
>
> 2. **[Standardization of the processing of multivariate data]** We have provided the preprocessing of the sensor data and processing of the activity annotations in the supplementary material (see Section 1.5 Preprocessing/cleaning/labeling). Moreover, the floor plans are vector images(svg) that represent the placement and adjacency of the sensors with respect to the room structure of an indoor space. We have uploaded these images in the github repo.
>
> 3. **[Missing data and Sensor calibration]** We have already provided the calibration details in the supplementary sheet (see Section 1.4 “Correctness”). We have also added information and additional discussion related to missing data in the Conclusion section of the paper.
>
> Please feel free to let us know if you have any further suggestions, comments or concerns. Thank you again for your time.

---

### Official Review · Reviewer_cXtY · 2024-07-24
**A good Indoor Air Pollution collection approach that could be better executed**

**Rating:** 6
**Confidence:** 3
**Clarity:** Yes

**Review:**

1.
I noticed an issue with regard to data cleaning. You claim that the data is smoothed with a 60-second rolling window to reduce noise. This is done at step 4 after imputing the missing values to 0. This is causing erroneous values at the start and end of real-data by smoothing them by the imputed 0s, making the processed data unusable.
This issue visible at timestamp 2023-06-10 12:33:25 (and others) @
https://media.githubusercontent.com/media/prasenjit52282/dalton-dataset/main/Processed/A1/2023_06_10/101_Study_Desk.csv .
This may impact the analysis derived with the data. *This issue needs to be fixed before taking the data for public release (irrespective of the ratings) and my rating already considers that you will handle this.* One approach is to smooth before imputing 0s for missing data.

2.
Fig 4, comparing the pollution traits for split and window air conditioners, do not show same time on x axis for both window and split ACs. There are 5 household sites and 5 research labs with both types of ACs.  It would be better to show the plots from same household/lab over the same duration so that the ambient temperature and pollution affects are neutralized.
Also AC on / off are the two coloured situations you show in the figures, then what does the white background signify? Is it due to missed annotations.

3.
A 25 weeks span covers 4200 in hours.
As per table 2 in appendix,
out of possible 4200 hours span, only two sites provide 2600+ hours data but without any annotations.
Other 28 sites contribute less then 850 hours each, with 7 sites even less than 90 hours.
So the possible pollution and annotation correlation is trivial or limited, which also impacts the usability of the dataset.

4.
The participants have sometimes provided multiple annotations in a single delayed communication.
eg. *2023-05-28 23:41:47," boiling rice approx. from 11:45 am to 12:15 pm. mutton curry approx. from 12:30 pm to 2:30 pm. water spinach sautÃ¨ing in oil from approx 2:00 pm to 2:30 pm. jackfruit, mango placed on dinning table from 4:45 pm to 10:45 pm approx. ",H7,P16.*
It appears that the authors have NOT processed the annotations by splitting/cleaning them and marking them separately at appropriate times. Having good annotations is a highlighting feature of this dataset and this responsibility cannot be passed to the users of this dataset to clean and use the annotations. Users will not have the same confidence in the participants like you have and hence cannot accept / discard the annotations in proper way. I would appreciate if the authors can alongside provide a processed annotation file, to best of their knowledge, which have valid and uniform annotations across participants and also do not contain merged annotations. This will reduce the usability barrier, and also will not allow different data interpretations in different bench-marking studies.

**Strengths:**

As discussed in the summary above.

**Additional Feedback:**

1. Phone number XXXX can be removed as it serve no purpose.

2. If possible, also provide the approximate spatial locations (latitude / longitude) of the different sites, so that the community can complement the given indoor pollution dataset with existing outdoor pollution data sources (like CAAQMS) and other meteorological parameters, increasing the dataset usability.

3. As you already have a good platform in place, I hope you will consider try deploying again in future with the suggestions provided by the reviewers, irrespective of the outcome of current dataset.

**Correctness:**

There are concerns with dataset cleaning, preprocessing and uniformity in plots as discussed above.

**Documentation:**

Sufficient dataset documentation is available.

**Ethics:**

Human attributes and privacy concerns seem limited.

**Limitations:**

Authors have already pointed missing annotations due to lack of participation by some of the participants. But the most contributing sites do not support and provide annotations, making the dataset usability limited.

**Opportunities For Improvement:**

1. The data availability is shown in the appendix in table 2.
I suggest to show the data availability plots with max 30 lines for important sites (one line for each site) having the horizontal axis denoting complete time span of summer and winter.
One plot shows the OR of all parameters availability. If any one device at a site is contributing data, consider data available for that site that hour.
Another plot shows the AND of all hourly data availability across all devices at each site, means complete data at a site is considered per hour.
Finally, a third plot with AND of the above with annotation. If no annotation in that hour on the site, consider the data is incomplete and hence unavailable. It may be similar to a simple annotation only plot.
This would clearly visualize the amount of data available across sites, and the potential dataset-users will appreciate before considering to use the dataset.

2.
There is claim of activity and annotations by the 46 participants, but as per appendix table 2, annotations are known to be missing for some participants. So, this claim may be softened.

3.
The authors acknowledge in limitations that participants fail to check their phones for notifications, miss annotations, or delay providing annotations beyond the specified timestamp. This may be partly due to missing motivation for timely action. The authors have a motivation to have a good research outcome, but ordinary participants with fixed remuneration may lack the enthusiasm. In my opinion, to promote active participation, a part of the remuneration should be kept variable and the participants be rewarded on providing timely and appropriate annotations.

4.
Also, the points mentioned in the Review section.

**Relation To Prior Work:**

Seems reasonable.

**Summary And Contributions:**

1. An indoor pollution dataset from diverse conditions like rural, semi-urban and urban from a developing country India, constructed with a sound approach.

2. It is a multi-device, multi-city, multi-pollutant, multi-week indoor pollution dataset with elaborated human annotations when and where available.

3. It also lists useful applications like Analysis of Spreading / Accumulation Patterns, Healthy Home Characterization, Smart Device Control, Pollution Source Identification and Activity Monitoring.

4. Some analysis on the Split vs Window air conditioners and the accumulation / spreading of the pollutants is discussed.

---

> ### Author Rebuttal · Authors · 2024-08-17
>
> Thanks for pointing out some important issues about the dataset. We have addressed the review comments and plan to update/ have updated the manuscript, the supplementary documents, as well as the online dataset repository as highlighted in the rebuttal summary. Here are the specific responses to your individual review comments.
>
> 1. **[Erroneous values]** We have addressed this by following your suggestion through smoothing the data and then replacing any missing values with zero in order to prevent any issue. We have updated the repository with the corrected data. We plan to update the supplementary material with the description of the corrected data.
>
> 2. **[Regarding Figure 4]** The figure already shows measurements from the same site; however, we understood your confusion and can update the figure (as attached) accordingly by keeping the time-scale (x-axis) consistent in both the subfigures and marking all the instances of AC on/off events (in the earlier version, we just marked one of those instances, and so the white background). To evaluate the impact of a window and a split AC on the environment, we used them separately one at a time. Additionally, in both scenarios, there were the same number of occupants, but all other ventilations such as windows, doors, etc. were closed during the experiment hours, so the exposure to pollutants and ambient temperature was similar in both scenarios, regardless of the number of occupants. The temperature range is different due to the cooling preset of the ACs that does not impact the degree of ventilation.
>
> 3. **[Annotation unavailability]** The reviewer has correctly pointed out the difficulty in collecting levels for all the dataset instances. The two sites that have 2600+ hours of data are the two households where we initially deployed our devices and are still collecting data continuously. Notably, although we do not have detailed activity annotations from these two households, we have other labels like the household locations, household floor plan and typical number of members at those households, etc. Such information is also useful to develop various downstream models and applications like understanding the spread of the pollutants from the kitchen to other rooms, impact of outdoor pollution on the indoor (by correlating and analyzing the data available from the public air quality monitoring systems, such as https://openweathermap.org/weathermap or https://www.windy.com/), etc. In addition, we have activity annotations for 4080 hours, totalling 3957 annotations of data, which can be used for downstream research based on correlation between activities and pollution generation. To make this clearer, we have updated the column header to “Activity Annot. (Hrs)” in Table 4 of the supplementary along with a column of other types of labeling, like floor plan and number of participants, etc.
>
> 4. **[Annotation usability]** Thank you so much for catching this important issue with our data. We agree that this directly pertains to the usability of the dataset in various practical application scenarios. Following the suggestion from the reviewer, we have cleaned the annotations as much as possible and updated the repository with a cleaned annotation file (Annotations_cleaned.csv) alongside the original annotation (Annotations.csv) received from the participants.
>
> 5. **[Additional details about data availability]** Following the suggestions from the reviewers, we have updated the primary manuscript, data repository and the supplementary file with the following details. (a) We plan to clearly mention that 24 among 46 occupants actively participated by providing activity and event annotations. (b) We have updated Table 4 of the supplementary material with hours of annotation per site. (c) We have removed the phone number field from the dataset. (d) We have provided the spatial locations (latitude, longitude) of the sites in the revised Table 4 in the supplementary material. The revised Table 4 is attached with the response.
>
> 6. **[Remuneration]** We agree that a good reward mechanism for experiments is necessary for timely and appropriate annotations. In our current deployment, we are providing remuneration weekly (instead of monthly) to keep the participant motivated for providing good quality annotations.
>
> Please feel free to let us know if you have any further suggestions, comments or concerns. Thank you again for your time.

---

> > ### Comment · Reviewer_cXtY · 2024-08-20
> >
> > I reviewed the changes to the smoothening process. Previously, there was an issue at the 59 second intervals after missing data gap, where the raw values from the devices were averaged with some 0s. Now, those raw values are simply replaced by 0s. This is though better than the previous case as the erroneous values are not present, the authors may look further to revive the missing values from the 59 seconds. Whatever the approach you choose, I would ask you to mention / acknowledge in your manuscript.
> >
> > Anyways, I appreciate your acknowledgement of the improvement points and taking necessary actions. I am raising my rating by 1 point.

---

> > > ### Author Rebuttal · Authors · 2024-08-21
> > >
> > > Thanks a lot for your response. Please note that we have collected the data at 1Hz sampling rate, indicating one sample per second. The issue at 59 second interval is because we are using a 60-second sliding window for denoising the data (considering per minute denoising); therefore, the first 59 seconds of the data in any of the measure sequences are the raw values from the sensors (including partial missing values in that window, if any). The data gets corrected from 60 seconds onwards. We can reduce this window size to minimize the data loss; however, there will be a tradeoff on the data denoising as the denoising filter will be less effective with a smaller sliding window size. We'll mention these details in the manuscript.

---

### Author Rebuttal · Authors · 2024-08-17

We sincerely thank all the reviewers for appreciating our work and highlighting some major issues with the dataset along with the suggestions to improve the paper. We have taken the suggestions very seriously and have already revised the data repository and can provide additional details in the paper and supplementary. While we have also provided detailed responses to the individual reviewers’ comments, here is a summary of the major updates that we have done or plan to do in the repository, paper and the supplementary material.

1. Following reviewer cXtY’s suggestion, we have updated the repository to handle the missing values correctly. We have also provided a cleaned annotation alongside the original annotations provided by the participants.

2. Following reviewer TNmg’s suggestion, we can provide additional benchmarking details in the revised supplementary material (details with the individual review rebuttal).

3. Following reviewer 2kUV’s suggestion, we can provide the details about the possible dataset applications (details with the individual review rebuttal).

4. Following reviewer 97sg’s suggestion, we can give some results from pilot studies in the supplementary material to highlight the possible utilities of the dataset in pollution sensing and other domains like food journaling (details with the individual review rebuttal).

Please check the detailed rebuttals for individual reviewers and provide us further suggestions, if any, to improve the paper. We again thank the reviewer for giving the detailed suggestions.

---

### Decision · Program_Chairs · 2024-09-26

**Decision:**

Accept (Poster)

**Comment:**

This paper proposes quite a unique dataset of indoor air pollution, including eight commonly occurring harmful pollutants (i.e., CO2 , VOC, PM1 , PM2.5 , PM10 , NO2 , C2H5OH, CO), across diverse indoor environments (kitchen, lab, canteen, studio, labs), from multiple sensor devices, across several cities in India.
The dataset is quite novel and diverse, with rich annotations that include indoor activities and events.
The authors are encouraged to improve the documentation of the dataset, and expand the details on the annotation processes to improve reproducibility, and discuss the applications and limitations of the datasets further.